# A large size-selective DNA nanopore with sensing applications

Rasmus P. Thomsen [1], Mette Galsgaard Malle[2,3], Anders Hauge Okholm[1,7], Swati Krishnan[4],
Søren S.-R. Bohr [2,3], Rasmus Schøler Sørensen [1], Oliver Ries[5], Stefan Vogel [5], Friedrich C. Simmel [4],
Nikos S. Hatzakis [2,3]* & Jørgen Kjems[1,6]*

Transmembrane nanostructures like ion channels and transporters perform key biological functions by controlling flow of molecules across lipid bilayers. Much work has gone into engineering artificial nanopores and applications in selective gating of molecules, label-free detection/sensing of biomolecules and DNA sequencing have shown promise. Here, we use DNA origami to create a synthetic 9 nm wide DNA nanopore, controlled by programmable, lipidated flaps and equipped with a size-selective gating system for the translocation of macromolecules. Successful assembly and insertion of the nanopore into lipid bilayers are validated by transmission electron microscopy (TEM), while selective translocation of cargo and the pore mechanosensitivity are studied using optical methods, including single-molecule, total internal reflection fluorescence (TIRF) microscopy. Size-specific cargo translocation and oligonucleotide-triggered opening of the pore are demonstrated showing that the DNA nanopore can function as a real-time detection system for external signals, offering potential for a variety of highly parallelized sensing applications.

[1] Interdisciplinary Nanoscience Center, Aarhus University, Aarhus C 8000, Denmark. [2] Department of Chemistry & Nanoscience Center, University of Copenhagen, Universitetsparken 5, Copenhagen 2100, Denmark. [3] Novo Nordisk Foundation Centre for Protein Research, University of Copenhagen, Blegdamsvej 3B, 2200 Copenhagen, Denmark. [4] Physics Department and ZNN/WSI, Technische Universität München, 85748 Garching, Germany. [5] Department of Physics, Chemistry and Pharmacy, University of Southern Denmark, Campusvej 55, 5230 Odense M, Denmark. [6] Department of Molecular Biology and Genetics, Aarhus University, Aarhus C 8000, Denmark. [7] Present address: Arla Innovation Centre, Agro Food Park 19, 8200 Aarhus N, Denmark. *email: hatzakis@nano.ku.dk; jk@mbg.au.dk

Lipid bilayers are key biological structures, serving as vital barriers for cells and subcellular organelles. To allow translocation of macromolecules and ions across this otherwise impermeable barrier, structural pores serve as transmembrane gatekeepers by creating hollow water-filled channels between the separated compartments. A large variety of protein pores exist in nature with various types of quaternary structure[1,2], and several successful examples of modified protein nanopores have driven the technological advancement within label-free biosensing and DNA-sequencing fields[3,4]. However, the lack of generic design rules for de novo protein design has limited its widespread application so far[4,5]. This is particularly evident for de novo-designed transmembrane proteins, where only recently computationally designed transmembrane proteins were successfully reported[6]. As an alternative, DNA has been established as a highly predictable building material for bottom-up de novo nanostructure creation[7], including DNA nanopores[4,8]. In particular, DNA origami[9] and single-stranded tile[10,11] techniques have excelled due to their extensive design space, while computer-aided software has further streamlined the rational design of complex three-dimensional DNA nanostructures[12]. With DNA origami, a single-stranded kilobase DNA scaffold is assembled into the designed structure by using hundreds of shorter staple strands that easily can be modified individually[13]. Thus, advancements of DNA modifications[13–15] and dynamic DNA structures[16] have enabled the construction of programmable and functional DNA-based nanomachines[17–20]. Of particular interest for this study are the biomimetic nucleic acid-based systems developed to manipulate lipid bilayers, including DNA nanopores[20–29], DNA-programmed SNARE mimics[30,31], and membrane-shaping structures[32,33].

Among the intrinsic problems of inserting a DNA nanopore into a lipid bilayer is its negatively charged phosphate backbone, creating an energy barrier known as the Born energy[34]. To enhance membrane association, DNA structures are usually decorated with hydrophobic moieties[35] by including lipid-modified DNA oligonucleotides (LiNAs) as staples or by capture handles.

Existing DNA nanopore structures have primarily been dominated by six-helix bundle (6hb) stem designs[20–23,28,29] and mainly focused on ionic current measurements and voltage-gating functionalities. Only recently, smaller[24] and larger[26,27] DNA nanopores have expanded the structural repertoire, demonstrating cytotoxicity[23], translocation of larger molecules[26], and charged-substrate or voltage gating[28,29] that have expanded the technological repertoire.

In this study, we create a rationally designed DNA nanopore with the largest channel lumen to date, which we have combined with a programmable trigger that significantly expands the functionalities of nanopores. With direct observation and in-depth analysis of the insertion and translocation kinetics of individual nanopores on liposomes, we analyze the functionality of the DNA nanopore and demonstrate a size-selective, modular and responsive gating of molecules between compartments separated by lipid bilayers.

The development of environment-responsive features, enabling autonomous structural actuation and cargo capture/release upon signal sensing, as demonstrated by other DNA devices[18,36,37], has the potential to transform the artificial nanopore field by enabling programmable insertion and biosensing of macromolecules.

## Results

**A dynamic and rigid DNA origami nanopore**. For the design of our synthetic DNA nanopore, we defined three important advances, illustrated in Fig. 1. (I) The pore is created from a double-layered pseudosymmetric hexagonal DNA structure with a 9-nm-wide lumen and an outer diameter of 22 nm, allowing translocation of large macromolecules including globular native proteins of more than 150 kDa in size (Fig. 1a). (II) The design contains numerous sites for functionalization in the channel interior, and (III) programmable DNA flaps that can be opened to expose lipid moieties. In the closed state, the flaps are locked by staple strands at the base to shield the hydrophobic moieties from the aqueous environment until activation (Fig. 1b), limiting hydrophobicity-driven aggregation[37]. When fully complementary key strands are presented by the liposome or added in solution to drive toehold-mediated strand displacement mechanism, the flaps are opened, and the exposure of lipids drives membrane insertion (Fig. 1d).

The three flaps were attached about 12 nm from the base of the 32-nm core channel by single-stranded DNA hinges and locked at the base by two staple strands. Importantly, when introducing these features, care was taken not to compromise the structural stability and introduce obvious kinetic folding traps (Supplementary Note 1 and Supplementary Fig. 1).

Based on the theoretical considerations of energetic penalty for inserting a pore of this channel size (Supplementary Note 2), a total of 46 lipidation sites were introduced on the channel and flap surfaces (Fig. 1d and Supplementary Fig. 3). The DNA pore was decorated with both cholesterol (18/46) and palmitoyl (28/46) (Supplementary Figs. 3 and 4) by including the synthesized LiNA staples into the self-assembly mixture. As expected, an increased association to liposomes is observed by increasing the number of hydrophobic moieties on the pore; thus we used the full lipidation scheme (Supplementary Fig. 5).

Using negative-stain transmission electron microscopy (nsTEM), raw images along with subsequent single-particle 2D analysis confirmed the correct folding of the DNA nanopore and function of the programmable flaps (Fig. 2). While measurement of the nanopore length from the 2D-class average estimated the length to be 35 nm, quantification from a limited set of cryoEM images provided a length of 31.6 nm, which is very close to the theoretically expected 32 nm (Supplementary Fig. 12). The lumen is clearly visible, both from the side and top views, and measured 9.6 nm at the widest dimension. From the top view the three flaps are visible as densities attached to the three larger hexagonal sites. In addition, a thermal denaturation assay confirmed high structural stability with a melting temperature of 56 °C (Supplementary Fig. 13). To enable hierarchical assembly of an elongated two-way pore, we designed two versions (A and B), each with 27 complementary sticky ends (Fig. 1c). Mixing A and B nanopores in solution resulted in a dimeric pore of the expected size, by aligning the lumens into a continuous elongated channel (Fig. 2 and Supplementary Figs. 6 and 7). Using this approach, dimer structures increased from 15% for noncomplementary A or B pores to a plateau of 70% for end-complementary A + B pores (Supplementary Fig. 14).

To confirm the dynamic properties of the flap structures, both nsTEM and FRET assays were used. The nsTEM analysis clearly showed extended protruding flap structures from the channel connected at the intended hinge region upon incubation with the key strand (Fig. 2c). Positioning of a Cy5–Cy3 FRET pair in a flap compartment further verified efficient and specific flap opening (Supplementary Fig. 15). The addition of the correct key sequence resulted in a 45% decline in the relative FRET signal within the first 5 min, whereas scrambled sequence did not have any effect. We conclude that the designed key strand can efficiently open the locks within minutes, leaving the flaps protruding from the core structure.

**Insertion of the DNA nanopore into lipid bilayers**. Next, we studied how the pore engages with bilipid membranes. Assembled

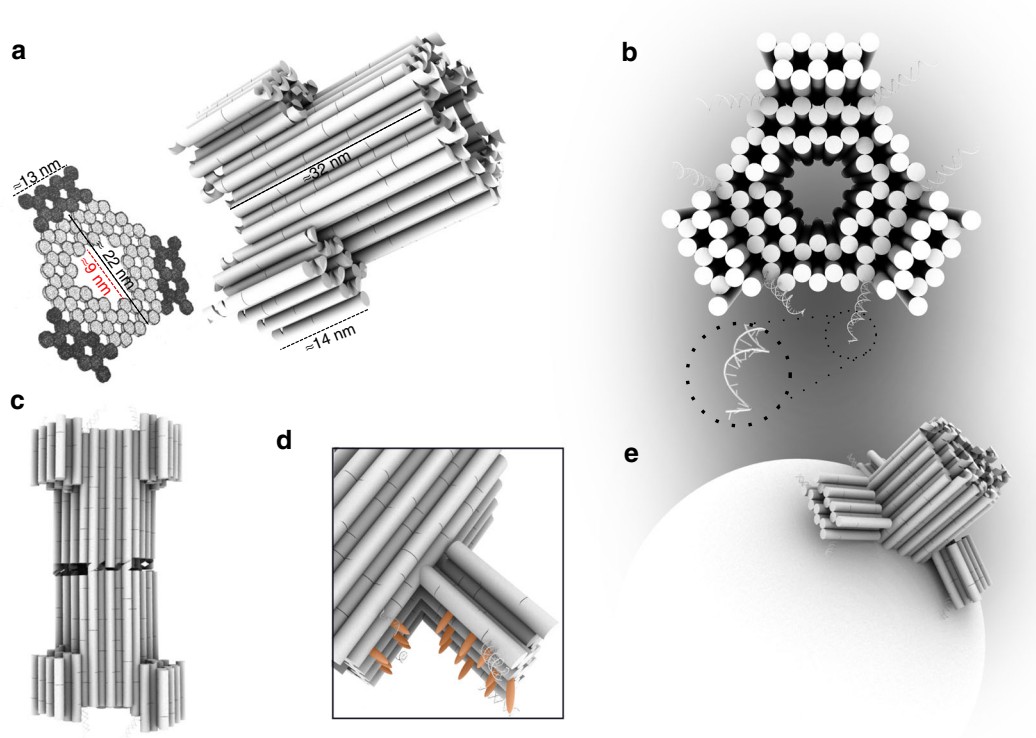

**Fig. 1 Design of the DNA nanopore. a** The pseudosymmetric nanopore is based on a hexagonal origami lattice and has a 9-nm inner pore diameter, a 22-nm outer diameter, and a length of 32 nm. **b** The three flaps are each locked with two dsDNA hybrids that possess an 8-nt toehold for strand displacement opening. **c** At the top, 27 staple strands are extended with unique 8-nt sequences to allow specific sticky end-mediated dimerization upon mixing A and B nanopores. **d** Lipidated nucleic acid staples (orange protrusions) are displayed at the surface of the channel and the flaps when opened. **e** Schematic illustration of the nanopore inserted into a liposome.

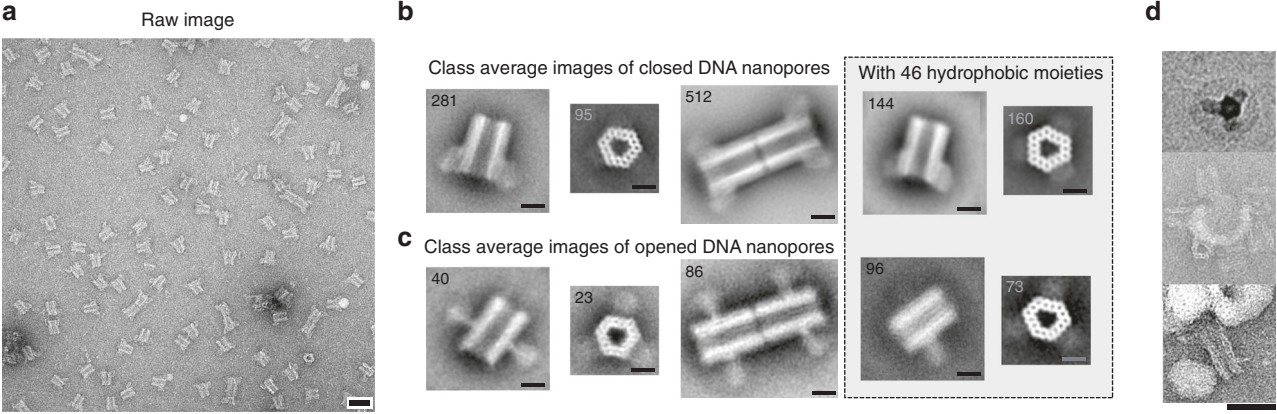

**Fig. 2 TEM characterization of the DNA nanopore. a** Raw nsTEM image of non-lipidated closed DNA nanopores. **b** Class-average images of the DNA nanopores in the closed- and **c** open-flap conformations. Grayed box shows closed and open nanopores assembled with 46 LiNA staples in the flap-channel system. **d** Raw TEM image cutouts of lipidated DNA nanopores incubated with liposomes. In the lower image, dimerized DNA nanopores have been formed prior to incubation with SUVs. Scale bars in (**a**) and (**d**) are 50 nm, while the scale bars in (**b**) and (**c**) are 15 nm. Additional raw images are provided in Supplementary Figs. 6–10. Sizes of SUVs have been analyzed by DLS (Supplementary Fig. 11).

nanopores were mixed with small unilamellar vesicles (SUVs) and visualized by nsTEM. Although not a quantitative assay per se, we observed many examples of flap-mediated insertion and interaction by using pre-opened nanopore structures, providing initial evidence of a functional DNA nanopore design (Fig. 2d and Supplementary Fig. 10). Interestingly, by dimerizing A and B nanopores, we were able to connect adjacent SUVs, suggesting a possibility to create channeling gates between vesicles/organelles, which would allow for gated translocation between compartments (Fig. 2d, lower).

To directly image nanopore docking on individual SUVs, pore formation, and to characterize dye translocation kinetically and

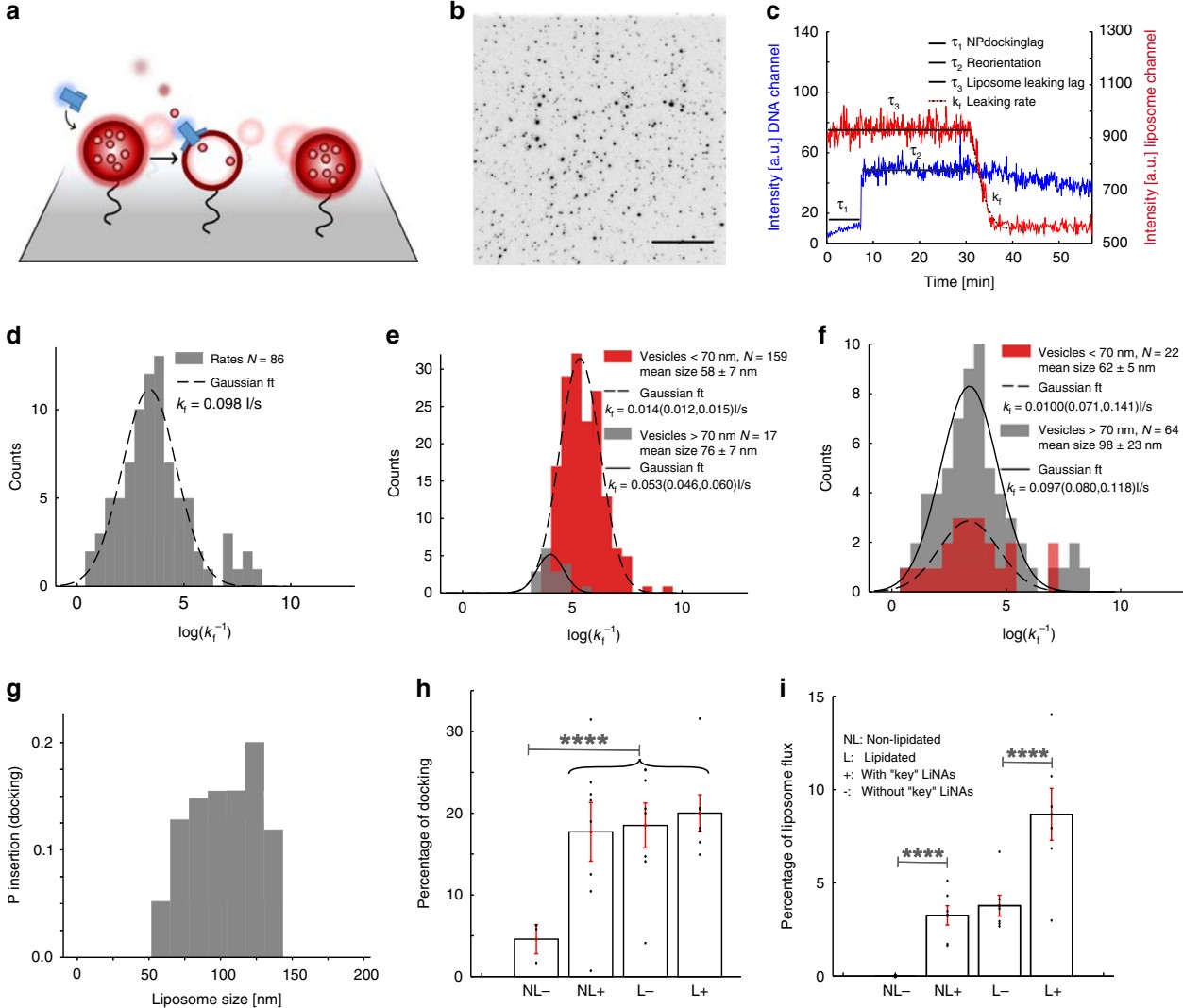

**Fig. 3 TIRF studies of DNA nanopores interaction with SUVs. a** TIRF setup, to monitor pre-opened lipidated DNA nanopores labeled with ATTO 488 used in (**c**), (**d**), (**f**), and (**g**) and interacting with ATTO 655-loaded SUVs tethered on PLL-PEG-passivated surfaces by a biotin–neutravidin linkage. Insertion of a DNA nanopore allows ATTO 655 efflux. **b** A single field of view images hundreds of surface-tethered ATTO 655-filled SUVs. **c** Representative traces of an individual nanopore docking (blue trace) on a SUV followed by dye leakage (red trace). The single-particle setup allows real-time observations of the docking time ($\tau_1$), reorientation time for pore insertion ($\tau_2$), as well as the ATTO 655 flow rate ($k_f$). **d** Histogram of flow rates from 86 recorded ATTO 655 efflux events fitted with a single Gaussian fit to extract the average flow rate. **e**, **f** All dye flow rates from SUVs observed separated by the SUV diameter into two populations ($d_{m1} > 70$ nm $> d_{m2}$) for either α-hemolysin (**e**) or DNA nanopore (**f**). While the α-hemolysin flux rate shows dependency of SUV size, the DNA nanopore flux rate does not. The α-hemolysin distribution consists of a total of 182 fully formed and leaking pores. **g** The probability for the DNA nanopore to form a pore upon docking into liposomes is plotted as a function of SUV size. Detailed data are found in Supplementary Fig. 21b. **h** Percentage of SUVs docked with non-lipidated or lipidated DNA nanopores (NL or L). The SUVs are either key- or no-key-decorated ($+$ or $-$) and the experiment is performed in a continuous nanopore flow setup (Supplementary Fig. 21a). **i** Percentage of the DNA nanopore-docked SUVs that successfully result in pore formation and dye flux performed in a continuous nanopore flow setup with the same legend as (**h**). Asterisks (****) in (**h**, **i**) indicate a $p$-value of $<10^{-7}$ and reflect a significant difference between the compared populations based on two-sample $t$ test, while error bars show the standard error of the mean. The sample size for (**h**, **i**) is $N = 709$ composed from four different experiments. Scale bar in (**b**) is 20 μm.

thermodynamically, we used total internal reflection fluorescence (TIRF) microscopy[38,39]. In this system, SUVs loaded with the dye ATTO 655 were surface-immobilized on poly(L-lysine)-poly (ethylene glycol) (PLL-PEG)-passivated surfaces by using a neutravidin–biotin capture, which maintains the vesicles' morphology[38] and essentially leakage-free membranes[40]. Freeze–thaw cycles (see the Methods section) ensure unilamellar SUVs as we showed recently[38]. Here, the immobilized SUVs serve as targets for either pre-opened lipidated DNA nanopores labeled with ten ATTO 488 fluorophore strands or α-hemolysin as a positive control (Fig. 3a, b). Parallel imaging of two emission channels

permitted synchronous imaging of SUVs and DNA nanopores and allowed the direct real-time observation of individual nanopore docking and subsequent stochastic insertion. Here, pore and SUV colocalizations were tracked, while the dye efflux rate caused by pore formation was quantified. From the real-time synchronous readout of ~10,000 liposomes for up to 8 h with 2.8 s of temporal resolution, a total of 86 nanopore insertion and pore formation events were observed (Fig. 3c). This allowed us to extract three main kinetic descriptors by using in-house developed single-particle tracking and analysis software (Supplementary Figs. 18 and 19, Supplementary Video 1 and Methods

section): the docking lag ($\tau_1$—time from incubation to SUV docking), the reorientation time ($\tau_2$—time from docking to pore formation), and dye flow rates ($k_f$) from perforated SUVs (Fig. 3c). Imaging optimization ensured minimal bleaching of ATTO 655 within the experimental time frame (Supplementary Fig. 20) and extraction of accurate efflux rates. The nanopore docking lag ($\tau_1$) follows a monoexponential decay process with a decay rate of 1450 s, suggesting a one-step process of nanopore–SUV interaction (Supplementary Fig. 21a). The reorientation phase on the other hand showed a complex insertion process (Supplementary Fig. 22). The prolonged time from nanopore docking to efflux, as well as the diverse efflux rates, support that the nanopore does not cause destruction of the vesicles upon docking. The flow rates were fitted by using a Gaussian distribution to extract the time derivative of the intensity, which revealed a rate of 0.098 Is$^{-1}$ through the DNA nanopore with a 95% confidence interval of (0.81,0.118) Is$^{-1}$ (Fig. 3d). In comparison, from $\alpha$-hemolysin control traces, a sevenfold lower average flow rate of 0.014 (0.013,0.016) Is$^{-1}$ was determined as expected due to the narrower pore size. Looking at the complex system by using a simplified theoretical approach, the flow rate and pore size relationship are found to be approximately following the Hagen–Poiseuille relation (see Supplementary Note 3 for theory and approximation). Splitting the observed flow rates into two populations, SUVs with a diameter ($d_m$) above and below 70 nm, enabled us to address membrane tension[41,42] on the pores and evaluate the relative mechanosensitivity of $\alpha$-hemolysin versus the DNA pore (Fig. 3e, f and Supplementary Table 1). In agreement with previous reports[41], the flow rate of $\alpha$-hemolysin appears mechanosensitive, exhibiting a flow rate of 0.014 (0.012,0.015) Is$^{-1}$ in small SUVs ($d_m < 70$ nm) versus 0.053 (0.046,0.060) Is$^{-1}$ in large SUVs ($d_m > 70$ nm) (Fig. 3e). Independent two-sample Kolmogorov–Smirnov test between the two distributions confirmed distinct populations (KS = 1, $P = 1.34 \times 10^{-6}$). In contrast, performing the same analysis with the DNA nanopore did not yield any difference in flow rate between the populations indicating a mechano-insensitive pore (Fig. 3f). Our DNA nanopore is capable of withstanding bilayer-mediated stress as opposed to previous measurements and simulations on single-walled 6hb structures[29,43], indicating the double-layered DNA pore structure to provide extra rigidity. Additional future experiments are required to verify if the observed mechanoinsensitivity further can suppress voltage gating, as observed with the single-walled 6hb structures. We conclude that our DNA nanopore has a rigid channel, which allows a monodisperse flow rate independent of membrane curvature.

Interestingly, while the liposome population is polydisperse in size and follows a log-normal distribution (Supplementary Figs. 20 and 28), the probability of successful DNA nanopore insertion upon docking has a slight preference for larger liposomes (Fig. 3g, Supplementary Fig. 21b and Supplementary Table 1). This occurs despite the fact that DNA nanopore and $\alpha$-hemolysin were exposed to identical SUVs. These findings agree with the qualitatively observed SUV insertion/interaction events from the nsTEM experiments (Supplementary Fig. 10), where insertion in smaller liposomes is rarely seen and indicates that the insertion process of the DNA nanopore is mechanosensitive. The reorientation phase on the other hand displays shorter lag times for smaller liposomes with higher curvatures (Supplementary Fig. 22) in agreement with earlier studies by us and others, demonstrating curvature-selective penetration of peptides and proteins[38,44]. Looking at the nanopore efficiency of insertion, about 20% of tracked liposomes were docked with DNA nanopores, from which about 11% was perforated during the experiment (Supplementary Fig. 21b). Although docking of $\alpha$-

hemolysin was not tracked, about 40% of liposomes were perforated by it. Considering that $\alpha$-hemolysin was used in a 15× higher concentration an approximately equal perforation per pore was obtained.

To test the programmability of DNA nanopore insertion into specific SUV populations decorated with flap-activating LiNA key strands, we conducted the following test: non-lipidated or lipidated DNA nanopores (NL or L) were subjected to a continuous flow over SUVs preincubated with or without (+ or −) LiNA key strands (Supplementary Fig. 23). As expected, non-lipidated nanopores showed practically no detectable bilayer docking on plain SUVs (NL−). Meanwhile, 17% of the observed SUVs were docked by nanopores upon decoration with LiNA key strands (NL+), demonstrating targeted docking only when the flaps were targeted (Fig. 3h). Using lipidated nanopores (L+), the docking lag was considerably reduced (Supplementary Fig. 23b), an effect that most likely arises from the increased lipidation. Interestingly, no significant difference in docking of lipidated nanopores on plain SUVs (L−) or key-decorated SUVs (L+) was observed, suggesting a tethering mechanism independent of flap activation for lipidated structures. Importantly, looking at the number of insertion events, defined by the beginning of dye efflux from the liposomes with an associated docked nanopore, we observed a 2.2-fold increase in events by using lipidated DNA nanopores on key-decorated SUVs (L+) compared with plain SUVs (L−). This indicates that exposure of the lipid groups enhances the fraction of docked pores that successfully insert into the membrane and result in dye efflux (Fig. 3i). Thus, effective DNA nanopore penetration can be controlled by surface-presented key signals, enabling individual addressability of SUVs.

**Nanopore channel size and size-selective gating of molecules.** To further evaluate membrane insertion and molecular transport through the channel, we conducted a dye-influx assay of surface-immobilized giant unilamellar vesicles (GUVs), as previously described in other nanopore studies[26] (Fig. 4 and Supplementary Fig. 25). Unlike SUVs, GUVs are experienced as flat lipid bilayers by nanostructures, thus providing a better mimic for physiological cell membranes. In order to track the insertion and channel accessibility, a combination of three differently sized fluorescent molecules were added to the external buffer solution prior to addition of Cy5-labeled DNA nanopores: a small (558 Da) sulforhodamine B (Rh ≈ 0.5 nm; SRB) combined with either a 40-(Rh ≈ 4.8 nm[45]; dFITC-40k) or a 500-kDa (Rh ≈ 15.9 nm[45]; dFITC-500k) dextran–FITC dye (Supplementary Fig. 25). Confocal laser scanning microscopy (CLSM) was used to visualize dye influx into the GUVs over 8 h. In line with the TIRF studies described above, only lipidated nanopores were inserted into the lipid membranes and this happened within the first 30 min. The SRB can readily pass through the DNA nanopore and is used as the positive readout of influx, and a total of 36 rapid (single frame = 10 min) SRB-filling events with intact membranes were registered and analyzed to gain a qualitative study from simple True/False answers. Slow multiframe filling events were interpreted as leaky bilayer influx and omitted (Supplementary Fig. 26).

As expected from the 9-nm lumen of the nanopore channel, the smaller dFITC-40k dye molecule was also able to translocate into the GUVs, albeit with a slower flux than SRB (Fig. 4a–d). As expected, the large dFITC-500k was unable to translocate across the membrane and only SRB influx was observed (Fig. 4e–h). This result demonstrates size-selective translocation of individual molecules with the expected cutoff value.

Next, we investigated the potential to control channel flux by introducing an addressable molecular plug. Up to ten 20-kDa

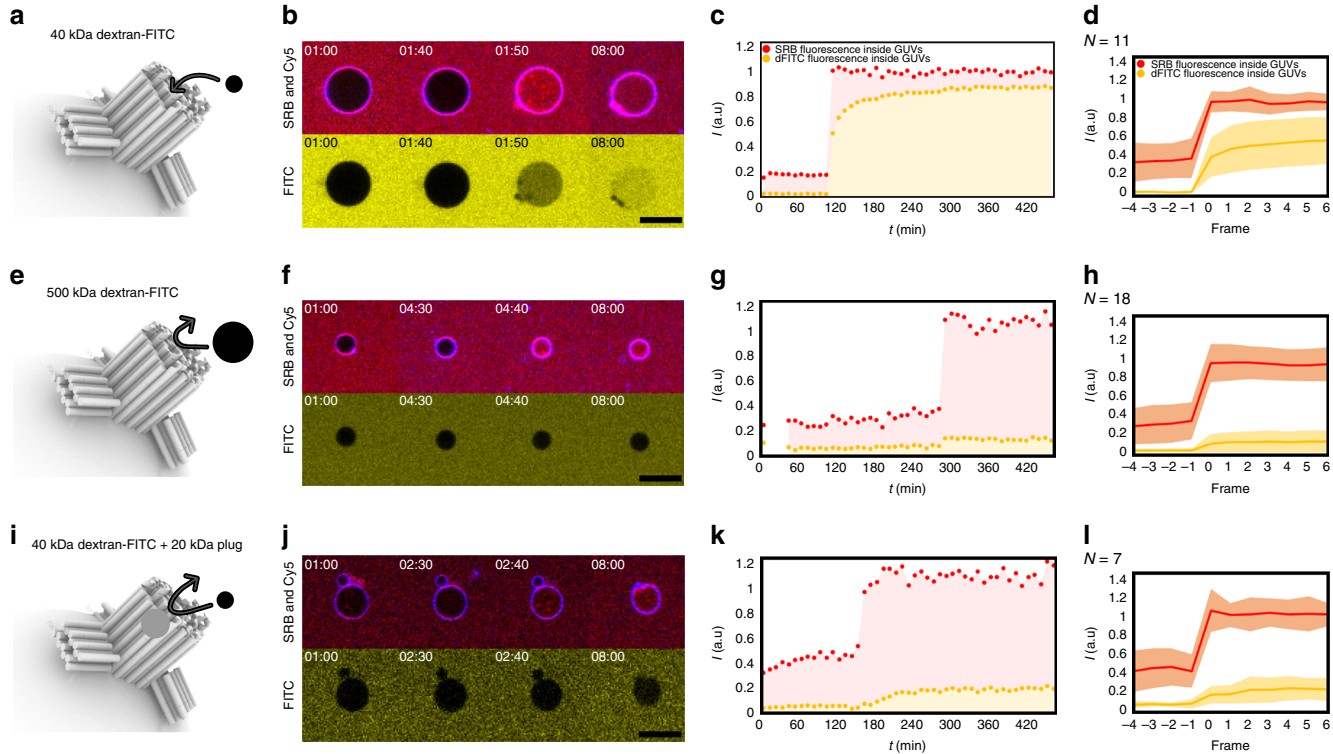

**Fig. 4 GUV dye-influx assay with DNA nanopores.** GUV experiments of opened lipidated nanopores with SRB and 40-kDa dextran–FITC (**a–d**) or SRB and 500-kDa dextran–FITC in outer solution (**e–h**). **i–l** GUV experiments with SRB and 40-kDa dextran–FITC in outer solution together with DNA nanopores containing 20-kDa PEG moieties immobilized in the inner channel as a size-selective plug. **a**, **e**, **i** Schematic of DNA nanopores penetrating lipid bilayer. **b**, **f**, **j** Time series of confocal images at the given intervals (hh:mm) of SRB (red), Cy5-labeled DNA nanopores (blue), and FITC (yellow). (**c**, **g**, **k**) Single traces of fluorescent signal from the inside of GUVs from $t = 00:00$ to $t = 08:00$ (hh:mm). **d**, **h**, **l** Average fluorescent traces of influx-normalized events for $N$-recorded events. All recorded rapid SRB influx events are included and data found in Supplementary Tables 2–4. The shaded regions surrounding the lines illustrate the standard deviations from the traces. Scale bars in (**b**, **f**, **j**) are 10 μm.

PEG polymers (Rh ≈ 4.9 nm[35]) were immobilized within the DNA pore by using available staple strand overhangs (Supplementary Figs. 16 and 17). As expected, this still allowed translocation of the small SRB dye into the GUVs. In contrast, dFITC-40k was now unable to pass through the PEG-plugged DNA nanopore (Fig. 4a–d), exhibiting a restricted influx similar to that seen for the large dFITC-500k (Fig. 4i–l). This demonstrates that plugging the channel can serve as a size-selective gate, distinguishing between cargo sizes of 15 and 5 nm.

To study the potential for the DNA nanopore to act as a real-time sensing device, we adapted the TIRF setup (see Methods section), performing a synchronous three-color imaging of ATTO 488-labeled nanopore insertion into ATTO 655- and 40-kDa dextran–tetramethylrhodamine (dTMR-40k)-loaded SUVs (Fig. 5a). Pore formation by a non-plugged DNA nanopore allowed full emptying of the SUVs as expected, supporting the existence of a large and rigid channel also observed from the CLSM influx assay (Supplementary Fig. 28). As a control, when using the narrow α-hemolysin pore, only ATTO 655 was able to translocate out of the SUVs whereas dTMR-40k remained encapsulated (Supplementary Fig. 29).

To introduce a controllable unplugging mechanism, we extended the PEG-tethering oligo with a toehold sequence of eight nucleotides, enabling PEG removal by a toehold-mediated strand displacement mechanism using a fully complementary DNA oligonucleotide unplugging strand (Supplementary Figs. 16 and 17). To examine the unplugging of the DNA nanopore, real-time, continuous imaging of DNA nanopores, ATTO 655 and dTMR-40k was performed for 9 h. DNA nanopore docking and pore formation into liposomes loaded with ATTO 655 and dTMR-40k resulted in selective initial efflux of the smaller ATTO 655. This in turn resulted in an increase of the TMR signal due to FRET dequenching (Supplementary Figs. 24 and 26). Unplugging strand was added after 4.5 h (dashed line in Fig. 5b) resulting in channel opening and the subsequent flow of dTMR-40k, demonstrating successful sensing capabilities of our nanopore (Supplementary Table 5). Interestingly, using the Stokes–Einstein diffusive model for the diffusive translocation of molecules indicates some hindrance for both dye molecules, probably due to interactions with the pore or hydrophobic interactions with the inner bilayer (Supplementary Note 4 for calculations and approximation). As nanopore docking on liposomes is a stochastic process that may take up to multiple hours, the experimental setup allowed us to sample events where insertion happens after addition of the unplugging strand, in which case the efflux of both dyes is observed simultaneously (Supplementary Fig. 27c). The differential flow rate of ATTO 655 and dTMR-40k and the observed dequenching of dTMR-40k upon ATTO 655 efflux, in addition, verify the integrity of the perforated SUVs.

In conclusion, our experiments demonstrate that controlled unplugging of the DNA nanopore combined with size-selective gating can be used as a sequence-specific sensing mechanism for external oligonucleotides.

## Discussion

We have successfully designed and prepared a DNA nanopore with a 9.6-nm pore inner diameter, which is significantly larger than existing DNA-based pores and demonstrated actuation-based insertion of the pore into GUVs and SUVs in response to specific keys. In addition, we have designed a plug system that

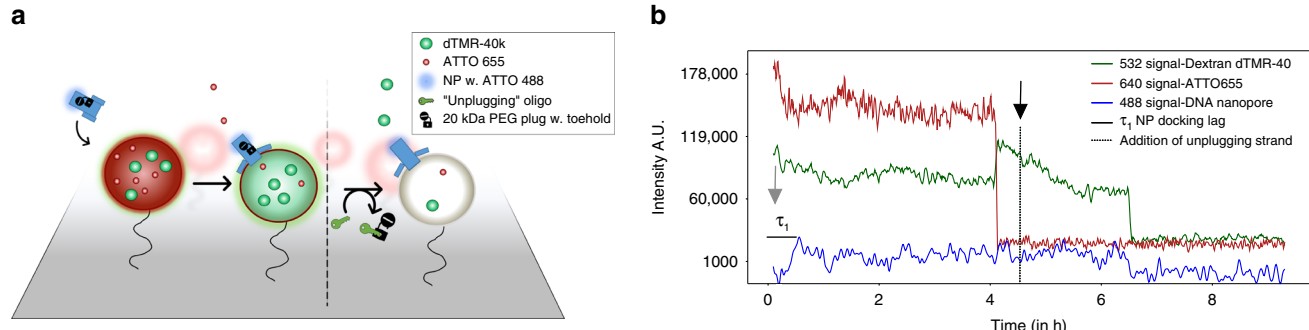

**Fig. 5 Real-time direct observation of sensing of an oligonucleotide unplugging strand. a** Schematic of three-colored microscopy setup. SUVs loaded with ATTO 655 and dTMR-40k are tethered on passivated surfaces. ATTO 488-labeled and plugged DNA nanopores are first added to solution in a flow cell. Upon insertion and pore formation, efflux of ATTO 655 through the channel-plugged nanopore happens, while the plugs restrict the translocation of the larger dTMR-40k dye polymer. Upon sensing the unplugging strand, the plug is released allowing passage for dTMR-40k dye to translocate out. **b** Single-particle trace of dTMR-40k- and ATTO 655-filled SUVs (green and red) docked by plugged DNA nanopore (blue). Before addition of the unplugging strand, insertion of the DNA nanopore permits only translocation of the ATTO 655 dye from the SUV. The gray arrow indicates nanopore flow in solution. After addition of the unplugging strand (black arrow and dashed line) and the subsequent actual unplugging event, the dTMR-40k will be able to translocate out. The observed increase in dTMR-40k fluorescence upon ATTO 655 flow is a result of dye dequenching and a proof of intact SUVs after insertion. All registered events are listed in Supplementary Table 5 and additional traces can be found in Supplementary Figs. 27 and 28.

allowed size-selective gating of translocation through the pore. Finally, we turned the DNA nanopore into a biosensor by demonstrating sequence-specific unplugging and dye efflux. We envision applications where other macromolecules are sensed by aptamer-based switches coupled to fluorescent detection. One of the advantages of the DNA pore is the possibility of spatially controlling virtually any modification in the lumen. For instance, one can imagine adding multiple plugs to the lumen to sense more complex, multicomponent signals or building enzymatic assembly lines.

Another application of the DNA nanopore to be explored further is its ability to create dimeric channels between adjacent lipid bilayers as previously also demonstrated by using engineered α-hemolysin pores[46]. In comparison, bifaced DNA pores can be constructed in a programmable fashion, thereby enabling controlled content mixing between liposomes.

Furthermore, our demonstration that the DNA nanopores can be targeted to prespecified SUVs, which can be addressed individually in a 2D TIRF setup, allows us to register thousands of molecular-sensing events simultaneously. This sets the stage for highly parallelized, label-free molecular sensing in the future.

## Methods

**Materials**. Unless otherwise stated, all DNA oligodeoxynucleotides, including the cholesteryl-modified versions, were acquired from IDT (Coralville, Iowa, USA). The ddT(hexanyl)TP was synthesized in-house. Ligations were done with recombinant Terminal Transferase acquired from Roche Applied Bioscience (Indianapolis, Indiana, USA). The azido-palmitoyl building block ((S)-1-azido-3-(palmitoyloxy)propan-2-yl sulfate **4**) was synthesized in 3 steps (Supplementary Note 5). In brief, commercially available (S)-Glycidol is acylated with palmitoyl chloride to give compound **2** in 95% yield which is subsequently turned into the corresponding azide by microwave assisted nucleophilic opening of epoxide **2** to provide azide **3** in 88% yield. Microwave assisted nucleophilic attack of the hydroxyl group on Sulphur trioxide followed by ion exchange with tetra-butylammonium hydroxide turns azide **3** into water soluble sulfate **4** in 87% yield. All fatty acids and liposome extruder parts were purchased from Avanti Polar lipids (Alabaster, Alabama, USA). Premade TEM grids were bought from TED-PELLA (Redding, California, USA). Uranyl formate was acquired from Polysciences Inc. (Warrington, Pennsylvania, USA) and kept in the dark. M13 was acquired from Bayou Biolabs (Metairie, Louisiana, USA) at 0.50 μg/ml concentration. Amicon Ultra-0.5 mL (100 MWCO) Centrifugal Filters for purification were bought from Meck Milipore (Darmstadt, Hessen, Germany).

SYBR Gold was purchased from Invitrogen (Thermo Fisher Scientific (Waltham, Massachusetts, USA)) and SYBR safe from Thermo Fisher Scientific (Waltham, Massachusetts, USA). All water was type 1 grade produced by MilliQ water purification system (or equivalent) unless stated otherwise and all solvents

were p.a. grade. Premade 10×TAE was acquired from Gibco (Thermo Fisher Scientific (Waltham, Massachusetts, USA)).

**DNA origami design and assembly**. The origami structure was designed as described by using CaDNAno, and the crossover pattern optimized through analysis of each staple with in-house written software. To create 3D models and renderings of the structure, Autodesk Maya (http://autodesk.com) with a CaDNAno plugin was used. Further, the 8-nt toeholds have been optimized against internal and unwanted structures with the NUPACK software. Modules of staples for the DNA origami were pooled from the plates acquired from IDT by our Eppendorf pipetting robot by using a custom-written software. Assembly of the structure was done by mixing a final concentration of 10 nM of scaffold with 5–10× excess of each staple strand (219 in total), 10 mM MgCl₂ (or 500 mM KCl if stated), and a 1×TAE buffer (40 mM Tris-acetate and 1 mM EDTA, pH 8.3), and diluted with MQ water to the final volume in a PCR tube. A heat-annealing program of 17 h was used. Initially a quick ramp from 90 to 65 °C was used followed by a slow ramp from 65 to 50 °C (−0.1 °C/20 min), and finally a last quick ramp to storage at 10 °C was programmed. Assembly of the origami with functionalities as Cy dyes and hydrophobic moieties was done similarly, by just replacing the non-functionalized oligos with the modified counterparts. DNA nanopore origami CaDNAno blueprint, information about the design and staple sequences can be found in Supplementary Fig. 31, Supplementary Note 6, and Supplementary Table 6.

Purification and concentration of the structures was done by 100-kDa MWCO Amicon ultra spin dialysis, prerinsed, and calibrated three times with a TAEMN buffer (40 mM Tris-acetate, 1 mM EDTA, 5 mM MgCl₂, and 5 mM NaCl) for 5 min at 10,000 × g. Afterward, TAEMN buffer and annealing reactions were added to the filter to a final volume of 500 μL and spun at 2000 × g for 15 min, and repeated 3–5 times. Elution of the retentate was done by inverting the filter and spinning at 1000 × g for 2 min followed by a quick wash of filters with 20 μl of buffer and eluted into the first eluate. Agarose gel electrophoresis was done with 1% agarose, 5 mM MgCl₂, and 6 μl of SBYR Safe dissolved in 1×TBE buffer unless stated otherwise. Identical concentrations of TBE and MgCl₂ were used as running buffer.

**Functionalized oligonucleotides**. The dideoxy(hexanyl)thymine triphosphate (ddT-hex-TP) synthesized in-house was enzymatically ligated with Terminal transferase to the 28 of the 46 lipid oligos (18 bought at IDT with cholesteryl). Ligation reactions consisted of a cacodylate (0.2 M) and Tris-HCl buffer (0.125 M, pH 6.6), CoCl₂ (5 mM) and BSA (0.25 mg/ml), Terminal Transferase (20 U/μl), DNA up to 80 μm (1.5 nmol), and 15× ddT(hex)TP in a total volume of 20 μl, incubated for a minimum of 6 h depending on DNA amounts at 37 °C and gently vortexed every 1 h for the first 3 h. The reactions were terminated by 20 mM final EDTA (pH 8.0) and EtOH precipitated by adding NaOAc (3 M, pH 5.6) and 2.5× volumes of cold EtOH. This was incubated on dry ice for at least 15 min before being pelleted at 17,000 × g at 4 °C. The supernatant was removed, and the pellet was resuspended in HEPES buffer (150 mM, pH 7.5) and prepared for subsequent alkyne reaction with click chemistry. The standard Cu(II)-catalyzed click reaction was done by mixing 10 μl of click buffer (1.33 mM CuSO₄, (H₂O), 2.64 mM TBTA (DMSO), 50 mM ascorbic acid (H₂O), and DMSO to 20 μl—in total 50% DMSO) with 6 μl of DNA-alkyne (1.5 nmol in 150 mM HEPES, pH 7.5), 100× lipidated azide (20 mM in DMSO), and 14 μl of DMSO to a final volume of 32 μl and 65% DMSO reacted for 8 h at 50 °C. EtOH was purified again and resuspended in 195 μl

of 0.1 M TEAA buffer. The reacted product was RP-HPLC purified with a Phenomenex Kinetex XB-C18 column (150 × 4.6) mm by using a TEAA/MeCN gradient buffer system (Buffer A: 50 mM TEAA, pH 7.0 and 5% MeCN, Buffer B: MeCN). Program: 5–100% in 20 min, 0.6 mL/min. Fractions containing the product were pooled, lyophilized, and dissolved in 1×TAE to a final concentration of 100 μM confirmed by UV absorption at 260 nm and ready for use. Similarly, biotin was attached by using either NHS–biotin with an amine oligo or using terminal transferase and a biotin-11-ddUTP (Jena Biosciences, Jena, Germany). For PEG functionalization of plug, NHS-PEG20k was reacted with the amine oligo and purified by using RP-HPLC.

**Flow cytometry.** Fluorescently labeled DNA nanopores were incubated with GUVs prepared by the inverted emulsion assay in the given time. Flow cytometry was done by using a Galios Flow Cytometer (Beckman Coulter, IN, USA) where 10,000 counts were recorded for the statistics. Following this gating was done by using relevant software.

**Transmission electron microscopy (TEM).** A 2% solution of uranyl formate was made in 20 mM KOH. Loading and staining of the DNA origamis onto TEM grids was done by first putting them in a glow discharger for 45 s followed by sample incubation for 1–1.5 min (depending on desired coverage) and dried by careful blotting. This was quickly followed by staining with freshly thawed aliquoted uranyl formate solution. Staining was done in two steps, a very quick 5 s step for washing purposes and a second step for about 20–30 s depending on desired staining. The solution was blotted and left to fully dry in air for a few minutes. Imaging was done by using a FEI Tecnai G2 Spirit electron microscope operated at 120 kV. Acquisition of images was done by using a bottom-mounted TVIPS CMOS 4k camera (TEM-cam-F416). Subsequent class- average images were made with the Scipion software package[47]. For cryoEM, DNA origamis were prepared and purified as stated above to a final concentration of 500 nM in TAEMN buffer. The samples were incubated for 4.5 s on glow-discharged Quantifoil R2/2 Cu200 grids (Electron Microscopy Sciences, Hatfield, USA) before being plunged or frozen in liquid ethane by using a Leica EM GP2 plunge freezer (Leica Microsystems, Wetzlar, Germany). Imaging was done on a 300 kV Titan Krios EM (FEI—Thermo Fisher Scientific, Hillsboro, USA) equipped with a Gatan K2 Summit direct detector (Gatan, Inc., Pleasanton, USA). Movies were collected by using 300 kV at 130k magnification and 1.336 e/Å².

**SUV formation for TEM.** POPC (1-palmitoyl-2-oleoylphosphatidylcholine) liposomes with a diameter of 100–400 nm was made by dissolving the lipids in EtOH that was evaporated by blowing $N_2$, depositing a lipid film on the sides of the tube. The lipid film was then dissolved in the preferred buffer to a 10 mg/ml solution, vortexed vigorously for 2 min to give a milky solution, and diluted to 2 mg/ml prior extrusion. Prior to extrusion, ten freeze–thaw cycles were performed by moving the emulsion from $LN_2$ to a water bath at RT. Extrusion was done with up to 1 ml of sample in the assembled Avanti extruder set (610000-1EA, Avanti Polar lipids Inc., Alabama, USA) with PC membrane filters containing 100–400-nm-sized pores depending on the desired diameter of the liposomes. Liposome samples were then passed back and forth a total of 13 times and size was checked by DLS (Zetasizer Nano-ZS, Malevern Instruments, Malvern, UK) analysis. For TEM purposes, a 1 μl final concentration of 0.2 mg/ml liposomes was mixed with 5 μl of purified DNA origami and 1 μl of 1 μM key oligos and incubated for at least 30 min.

**FRET measurement.** FRET analysis was done by assembly of the structure with Cy3 and Cy5 dyes and the opening analyzed by measuring the emission of the ensemble on a Fluoromax 4 (Horiba Scientific, Kyoto, Japan), and calculating the relative FRET efficiency for qualitative analysis. Addition of oligos was done by opening the lid between measurements and closing it up before a new measurement.

**Inverted emulsion confocal assay.** Egg PC (770.123 g/mol) and biotinylated PE (941.95 g/mol) were dissolved in chloroform at 10 mg/ml and mixed in a 10:1 ratio, and diluted to 5 mg/ml. The chloroform was evaporated in a vacuum rotator for 1 h and left O/N in a desiccator to dry. The lipids were resuspended in mineral oil (Sigma) and vortexed well before 80 °C by heating in a water bath followed by vortexing and repeated ten times. Afterward, the solution was set in a sonication bath for 90 min at 60 °C. Fifty microliters of sonicated liposome solution was added to 150 μl of mineral oil and left for 30 min on ice. Twenty microliters of inner buffer (1×TAE + 500 mM KCl + 400 mM sucrose) was added and immediately emulsified by vortexing for 30 s to give a milky solution and incubated on ice for 30 min. Hundred and fifty microliters of liposome solution was placed in 1 ml of outer solution buffer (TAE + 500 mM KCl + 400 mM glucose)—important, osmolality difference must only vary by 20 (Gonotec Osmomat 010, Gonotech GmbH, Berlin, Germany). The solution was then incubated on ice for 30 min before centrifugation at 12,000 × g for 30 min at 4 °C. The oil layer and the uppermost top layer was removed; the pellet was resuspended and incubated on ice for 30 min. Slides for confocal assay were prepared by first pacification of the surface with BSA (5 μl of 1% BSA) before adding 55 μl of liposome mixture incubated for 30 min to sink. Atto633 or SRB dye was added and mixed followed by adding 15 μl of sample (Amicon purified and buffer exchanged to 1×TAE +

500 mM KCl) incubation just before confocal assay. This buffer did not change the osmolality significantly. The confocal recordings were done for 7–10 h with images every 5–10 min of multiple locations and heights.

**Acquisition of TIRF data.** All single-particle experiments were performed by using an inverted total internal reflection fluorescence microscope (TIRF) model IX83 from Olympus. The microscope was equipped with an EMCCD camera model imagEM X2 from Hamamatsu and a ×100 oil immersion objective model UAPON 100XOTIRF from Olympus and an emission quad band filter cube, in order to block out laser light in the emission pathway. Fluorophores were excited by using three solid-state laser lines from Olympus at 488, 532, and 640 nm in order to excite DNA nanopore, dTMR-40k, and Atto655, respectively. All data were measured with a 200-nm penetration depth and an EM gain at 300 and with image dimensions of 512 × 512 pixels with a dynamic range of 16-bit grayscale. The field of view corresponds to a physical field-of-view length of 81.92 μm.

**TIRF single-particle assay.** SUVs from DOPC lipids with 1% DOPS charges and 0.5% biotinylated lipids were subjected to ten cycles of flash-freezing and thawing to ensure a unilamellar structure. Afterward, the SUVs were extruded at 200 nm as described before in 400 mM sucrose and 200 mM KCl. The DNA nanopores demonstrated to be stable for >24 h when diluted in the TIRF solution (Supplementary Fig. 24). ATTO 655 was encapsulated following recently published methodology[40,48]. Glass slides with attached sticky-Slide VI 0.4 from Ibidi were functionalized with PLL-g-PEG and PLL-g-PEG–biotin in a 100:1 ratio and consequently covered by a neutravidin layer[39]. Biotinylated SUVs were flown into the system by using a peristaltic pump and left to immobilize for 10 min. Tuning of vesicle amount and equilibration time allowed to achieve vesicle densities of ≈300 vesicles per field of view. Flowing of buffer removed freely diffusing vesicles.

Long-term DNA nanopore measurements were initiated by flowing into the microscope slide ATTO 488-labeled 1.4 pmol opened DNA nanopores that were freshly prepared (about 20–30 μl of <40 nM purified DNA nanopore in 1×TAE, 5 mM MgCl₂, and 5 mM NaCl was mixed in 200 μl of 400 mM sucrose and 200 mM KCl flown in). Using α-hemolysin at 45 nM, the final heptameric pore was flown into the flow cell in identical buffer. For key experiments, lipidated key oligos were first flown into the system, and settled for about 30 min before rinsing as we have done lipidated key oligos recently for lipidated proteins[38] and followed by slow incubation of DNA nanopores at a flow rate of 10 μl/min with a total flow of 500 μl in 50 min. To increase throughput, we designed experiments where four fields of view were automatically imaged sequentially. In a typical experiment, the first field of view is recorded in all relevant imaging channels with exposure time of 100 ms, followed by the second, third, and fourth fields of view. When the cycle was completed, the automated software would return to the first position and initiate the cycle again. The temporal resolution for each image was 100 ms followed by a 500-ms change time, resulting in a final temporal resolution of 2.8 s for each cycle. A typical experiment lasted for 8 h.

**TIRF real-time sensing of unplugging assay.** SUVs were prepared as described in the previous section. Both ATTO 655 and 40-kDa dextran–tetramethylrhodamine (dTMR-40k) were encapsulated. Immobilized SUVs on the PLL-g-PEG and PLL-g-PEG–biotin-functionalized coverslip were washed thoroughly with 10 mL of 400 mM sucrose and 200 mM KCl buffer, in order to remove excess ATTO 655 and dTMR-40k that is either in solution or externally bound. Images were acquired for 100 ms for each of the three channels, by using the three laser lines sequentially, with six positions and a 500-ms change time, providing a temporal resolution of 4.6 s between each cycle. The assays were measured for 9.45 h with a continuous flow at 10 μl/min, to avoid outflowing dextran from sticking to the outer membrane.

**Tracking and localization software of the TIRF experiments.** Due to the extremely long experimental time frame (>9 h), an inevitable stage drift of liposomes was observed. To circumvent this problem, we deployed single-particle tracking by using the open-source ImageJ plugin TrackMate[49], to actively track and follow the drift of liposomes on the surface. This was done by localizing each individual liposome, in each frame for a given experiment, by using a Laplacian of Gaussian approximation. Hereafter the localized dots are connected through frames, by using Linear Assignment Problem Tracker (LAP Tracker)[50], thus creating time-resolved trajectories circumventing state drift. By applying the same methodology to the nanopore-imaging channel and colocalizing with liposomes, we were able to extract docking times and time-resolved intensity traces.

For analysis of three-color experiments (using encapsulated ATTO 655, dTMR-40k, and ATTO 488-labeled nanopores), it was necessary to extend and develop the extraction of signal. Due to increased noise from adding a third fluorophore to the system, the tracking algorithm described above failed to capture some events. We therefore developed a custom-made script, by utilizing the open-source particle-tracking python plugin TrackPy[51,52], to correct for stage drift. By simultaneously tracking all liposomes in the field of view, throughout each individual experiment, we were able to extract the average displacement (in x and y directions, respectively) between each consecutive frame (Supplementary Video 1). Using this information, we were able to correct for stage drift. This was done by

inserting each individual frame (512-by-512 pixels) of a given experiment into a bigger, empty frame (562- by-562 pixels) and then moving it according to the average drift (19 for detailed explanation). After drift correction, the signal for each channel would be extracted by using an in-house-developed routine that localizes spots on a surface using TrackPy and subsequently collects the signal from each spot and corrects for background noise.

**Reporting summary**. Further information on research design is available in the Nature Research Reporting Summary linked to this article.

## Data availability

The authors declare that the data supporting the findings of this study are available within the paper and its Supplementary information files. Additional and relevant data are available from the corresponding authors on reasonable request.

## Code availability

Codes used for tracking and colocalization in the TIRF setup are available from the Hatzakis group homepage, http://www.hatzakislab.com.

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

## Acknowledgements

We thank Thomas Boesen for the guidance of CryoEM. We thank Per Hedegård for the guidance with the diffusive model calculations. We thank Anne F. Nielsen for reading and commenting on the manuscript. This work was funded by the Villum foundation by being part of BioNEC (grant 18333) for R.P.T., M.G.M., O.R., S.V., J.K., and N.S.H., the Danish National Research Foundation for J.K. (DNRF135), the Villum foundation young investigator fellowship (grant 10099), and the Carlsberg foundation Distinguished

Associate professor program (CF16-0797) for N.S.H., and by the Deutsche For-schungsgemeinschaft through SFB 863/TP A8 for S.K. and F.C.S. N.S.H. is a member of the Integrative Structural Biology Cluster (ISBUC) at the University of Copenhagen.

## Author contributions

R.P.T., R.S.S., and J.K. conceived the project and planned the research. R.P.T. and R.S.S. designed the DNA nanopore and R.P.T. did all DNA functionalization and performed the TEM characterizations. R.P.T. and A.O. did the flow cytometry experiments and FRET studies. R.P.T., A.O., and S.K. set up and executed the CLSM GUV studies under supervision from F.C.S. TIRF measurements were planned by M.G.M. and R.P.T. under the guidance of N.S.H. and done primarily by M.G.M. with help of R.P.T. The TIRF data treatment was done by M.G.M., S.S-R.B., and N.S.H. O.R and S.V. provided lipid building blocks for the DNA nanopore functionalization. The paper was written by R.P.T. with inputs from all authors.

## Competing interests

The authors declare no competing interests.
