## [Peer Review File · Nature Communications]

Reviewers' Comments:

Reviewer #1:

Remarks to the Author:

The present manuscript by Thompson et al. presents a new membrane-inserting DNA nanopore with the largest pore diameter to date – potentially more suitable for single-molecule sensing applications compared to previous designs.

New design features include a double-layer of DNA in the membrane-penetrating part (for increased stability) and the incorporation of DNA flaps that shield the hydrophobic moieties and hence prevent aggregation. While previous DNA nanopores have mainly been characterized with ionic current recordings and confocal microscopy, the authors provide new insights based on TIRF microscopy. This includes a compelling quantification of docking and insertion kinetics as well as a method to measure the mechanosensitivity of the pores.

However, some questions remain open as outlined in my comments. This especially concerns the unilamellarity of the SUVs used for the characterization of the insertion characteristics and the lack of discussion of the mechanosensitivity. If the following points can be addressed, I recommend the manuscript for publication in Nature Communications.

Comments:

1. p. 2, l. 48: „Existing DNA nanopore structures [...] mainly focused on static ion conductance“. This statement is misleading. On the contrary, all DNA nanopores have been shown to exhibit voltage gating – to date a structure with static conductance across a broad voltage range has not yet been demonstrated, see Seifert et al. (<https://pubs.acs.org/doi/10.1021/nn5039433>) The statement has to be rewritten to make it less misleading.
2. The DNA flaps are actuated by toehold-mediated strand displacement. Why did the authors not simply rely on hydrophobic actuation as proposed by List et al. (Angewandte Chemie, 2014, DOI: 10.1002/anie.201310259)?
3. As hydrophobic tags, the authors use a mixture of cholesterol- and palmitoyl-functionalized oligos. What is the reasoning behind this? Have the authors quantified the benefits (less aggregation, better insertion...)? In the SI (Note S2), the authors discuss the free energy gain for pore insertion, but they just assume the same energy gain for cholesterol and palmitoyl insertion. This makes it even more unclear why they chose the mixture. This should be discussed.
4. Why does the nanopore appear longer in TEM than the theoretically expected value (35nm vs. 32nm)? Is it due to the staining?
5. p. 5, l. 83: “In addition, a thermal denaturation assay confirmed the high structural stability (Fig. S10)“ It would be helpful if the authors could add the determined melting temperature to this statement (56°C). This value is quite typical for DNA origami structures.
6. The dimerization is a nice feature of the DNA origami nanopores. How many percent of the pores are dimers before and after the introduction of the sticky ends? This information should be added to the main text. Why did they not attempt to measure the gated transport between compartments? Are the insertion rates too low?
7. The quantification of the insertion characteristics is very valuable. However, the conclusions are only valid, if the authors are certain that their SUVs are unilamellar. Have the authors quantified the fraction of multilamellar vesicles? I am concerned that it might actually be quite large, since extrusion was performed with large filter sizes between 100 and 400nm and without freeze drying.
8. The authors state that they observed 86 DNA nanopore insertions over the course of 8h. How many insertions of alpha-hemolysin did they observe in the same time period?
9. The authors look at “the smallest or the largest SUVs” (p. 7, l. 126) to study the mechanosensitivity of the pores. How small is small and how large is large and what is the difference in curvature? 100 nm vs. 400 nm? Can they estimate the difference in membrane tension?
10. The lack of mechanosensitivity of the DNA pores is a very interesting result, however it

contradicts previously published work: MD simulations showed that the conductance of DNA nanopores depends on membrane tension (Yoo et al., J. Phys. Chem. Lett., <https://pubs.acs.org/doi/10.1021/acs.jpcclett.5b01964>) and experiments showed that the conductance states can be influenced by membrane tension (Seifert et al., ACS Nano, <https://pubs.acs.org/doi/10.1021/nn5039433>). These results have to be mentioned and discussed! Do the authors believe the double layer of DNA is the reason? It would be really good if they could test the mechanosensitivity of previous pore designs in a follow-up study.

11. Osmotic pressure differences and ionic conditions in the buffer are crucial parameters governing the insertion properties of DNA pores. While the authors provide the buffer conditions for the formation of SUVs, GUVs and DNA origami individually, it is difficult to find out what final buffer was inside and outside the vesicles during the experiment. Providing this information explicitly will add value, especially for the quantification of the insertion characteristics.

12. The authors state that they made SUVs between 100 and 400nm by extrusion and checked the size via DLS (see Materials and Methods). However, the DLS data is not provided. This data should be added to the SI, especially since extrusion with larger filters often leads to a deviation from the desired pore size.

13. All previous DNA nanopores exhibited voltage gating, which somewhat limits their applicability for nanopore-based single-molecule detection. Do the authors believe that the use of a double layer of DNA would suppress voltage gating? This would be an immense advantage of the design. Ionic current recordings could provide the ultimate proof, but at least some discussion about this point should be added to the manuscript.

14. p. 12, l. 210: For clarity, replace "lumen size" by "pore diameter".

Kerstin Göpfrich

Reviewer #2:

Remarks to the Author:

The manuscript from Kjems and coworkers describes a synthetic DNA nanopore for the programmable and size-selective translocation of molecules through lipid bilayers.

The main novelty of this work relies on the functional versatility of the DNA nanopore structure. Differently from previously described DNA-based nanopores, this structure enables the size-selective (in and out) translocation of large or small molecules through modification of its lumen with PEG-ylated strands and their further removal by single-strand displacement mechanisms. In addition, the DNA nanopore inserts into lipid bilayers upon actuation by external triggers. These are DNA strands, either added in solution or present on the lipid surface, that bind to specific regions of the structure and operate by reconfiguring the nanopore from a closed to an open form, in which three flaps expose previously hidden lipid moieties. This structural change enables tight binding of the nanopore to the lipid membrane and avoids unwanted (and previously observed) hydrophobic-driven aggregation. The DNA nanopore and its functioning has been thoroughly characterized by TEM and smTIRF as well as by gel electrophoresis and FRET spectroscopy.

In general, the work is well written, clear and pleasant to read. I noticed only few typos (listed at the end of this report). I particularly liked the experiments described in figure 4 and 5. In my opinion, the controllable size-selective channeling properties of the DNA nanopore by plugging/unplugging of its lumen is the strongest point of this work and nicely shows how the smart combination of simple molecular mechanisms on robust scaffolds can be used to generate functional nanodevices with advanced properties. The data obtained are solid, consistent with the hypotheses and well-illustrated. I am therefore fully favorable to publication of this work in Nature Communication upon few minor alterations that I suggest below. The authors may consider to address my comments for improvement of the paper and a few other questions that could satisfy the curiosity of the interested reader.

1. Lines 59-65 describe the novelty and strength of the work. I like very much point III (actuation

of the device); however, I found somehow difficult to grasp it from the present description in the main text or from figure 1. This point became clearer to me only after re-reading how the device works, which is reported few lines below. I would therefore suggest to describe first the two states of the device (i.e. the closed and open form), making also the mechanical transformation from one to the other clearer in the schematics of figure 1. In my opinion, this will emphasize the smart strategy employed by the authors to control membrane insertion and will guide the reader smoother to the next points.

2. Which type of DNA nanopores has been used in the experiments illustrated in Figure 3a-g? maybe pre-opened L-? I find hard to guess it from the schematics in a). Can the authors please state it clearly in the main text and legend? What is the average size of a "small" SUV and a "large" SUV?
3. It would be interesting to see whether the numerical relationship between the flow rate and the pore size follows some particular law of diffusion. Do the authors think it may be useful to gather some more insights into the DNA nanopore functioning?
4. Line 152. "...suggesting a tethering mechanism independent of flap activation for lipidated structures". From the FRET experiments of Fig. S12, it looks like the lipidated structures are partially opened even in the "closed" configuration, which would actually well explain the high insertion yield of L- constructs. Can the authors shortly comment on that?
5. What is the exact difference between Figure 3h and 3i? Does it mean that there could be docking without dye efflux?
6. Experiments of Figure 4: did the authors check whether the obtained results of flow rate vs. particle size are in line with the expected values from the Stokes-Einstein equation on molecular diffusion? In general, I believe that by quantifying the observed effect in relation to a known physical law would give a stronger impact to this nice work and would possibly help to gain additional information on the performance of the synthetic device (see also point 3 above).
7. Experiments described in Figure 4 and 5. Plugging experiments were performed to monitor the different influx rates of dyes through GUV-tethered nanopores, whereas unplugging experiments were performed to monitor the efflux of previously GUV-internalized dyes. Is there a practical reason for this experimental choice? Do the authors believe that the performance of the device is affected by the direction of dye flux? That is, are influx and efflux rates identical for the same molecule through the same DNA nanopore?

Some comments on the Suppl. Information:

In general, at some points of the SI, there seems to be a formatting problem (some images are moved away from the legend and/or only partly visible).

8. The analysis of structural design is very interesting and will surely be of help for the development of the DNA nanotechnology field. Can the authors state which criteria must be fulfilled to have what they describe as a structural "valley"?
9. Also, the note on the free energy of insertion is very interesting. Here, I would suggest to rephrase the second sentence (apparently a typos) and define the values of " γ "_L and " γ "_S in the equation (probably the bilayer tension due to different contributions). The referred equation 4.2, for calculation of the number of hydrophobic tags needed, is not reported. It would be nice to have it directly in the SI to help the reader to grasp the message without recurring to literature search. A final point that the authors may consider to address is the energetic contribution given by palmitoyl when compared to cholesterol. Is there a particular reason why they decided to use 28/46 palmitoyl and 18/46 cholesterol moieties for DNA nanopore modification? As they are chemically very different (long hydrophobic chain vs. steroid), one would imagine that their energetic contribution to insertion is also different.
10. Fig. S20, legend: I believe left and right of panel a) have been exchanged.

Some typos and few minor comments:

11. Line 62: "the design permitS".
12. Figure 2d, lower: I am not able to see well the dimerized channel building a bridge between two vesicles. As this construct was not further analyzed in this paper, I would suggest the authors

to consider moving this TEM image to the SI.

13. Line 106: PLL-PEG, please expand abbreviation for PLL

14. Line 112: "...the dye efflux rate caused by pore formation was quantified". Is there the possibility that the intensity of the loaded dye may decrease by photobleaching? Did the authors check for that?

15. Line 114: " From ...10,000 liposomes a total of 86 nanopore insertion events was observed". The reader would conclude that the yield of insertion is <1%. From the TEM images reported in the main manuscript and SI, it seems to me that the yield of insertion events is much higher. Very often one SUV is bound to many DNA nanopores. How do the authors comment on that?

16. Figure 3: in the legend it should be stated that the DNA nanopore is labelled with ATTO 488.

17. Line 179: "... still allowed translocation of the small SRB dye out of the GUVs". Do the authors mean instead "into the GUVs"? All experiments described in this section refer indeed to dye-influx.

18. Figure 5: the final part of the legend is missing.

19. Line 128: "a-hemolysin appearS mechanosensitive ..."

20. Line 134: "DNA nanopore has a rigid channel, which allowS ..."

21. Line 163: can the authors provide the estimated R_h for the SRB? this will help the reader to make a full comparison between flux rates and molecular size.

Reviewers' comments:

Reviewer #1 (Remarks to the Author):

The present manuscript by Thompson et al. presents a new membrane-inserting DNA nanopore with the largest pore diameter to date – potentially more suitable for single-molecule sensing applications compared to previous designs.

New design features include a double-layer of DNA in the membrane-penetrating part (for increased stability) and the incorporation of DNA flaps that shield the hydrophobic moieties and hence prevent aggregation. While previous DNA nanopores have mainly been characterized with ionic current recordings and confocal microscopy, the authors provide new insights based on TIRF microscopy. This includes a compelling quantification of docking and insertion kinetics as well as a method to measure the mechanosensitivity of the pores.

However, some questions remain open as outlined in my comments. This especially concerns the unilamellarity of the SUVs used for the characterization of the insertion characteristics and the lack of discussion of the mechanosensitivity. If the following points can be addressed, I recommend the manuscript for publication in Nature Communications.

We thank the reviewer for acknowledging the importance of the manuscript the compelling quantifications and for recommending publication to Nature communications after addressing a few points. We have detailed our answer and highlighted changes in the manuscript main text and SI below addressing all reviewers comments.

Comments: **ORANGE edits in paper**

1. p. 2, li. 48: „Existing DNA nanopore structures [...] mainly focused on static ion conductance“. This statement is misleading. On the contrary, all DNA nanopores have been shown to exhibit voltage gating – to date a structure with static conductance across a broad voltage range has not yet been demonstrated, see Seifert et al. (<https://pubs.acs.org/doi/10.1021/nn5039433>) The statement has to be rewritten to make it less misleading.

- *We agree with the reviewer that voltage gating in more than one state has been reported. However, our intention with this sentence was to emphasize that most of the 6hb-like pores in the literature were tested with voltage gating but not using a molecular pore regulation mechanism. In the Seifert et al. paper, the two-state gating is done using an external voltage switching control. We have tried to clarify this distinction in the revised manuscript. See main article page 2, line 47-49.*

2. The DNA flaps are actuated by toehold-mediated strand displacement. Why did the authors not simply rely on hydrophobic actuation as proposed by List et al. (Angewandte Chemie, 2014, DOI: 10.1002/anie.201310259)?

- *Thank you for this interesting suggestion. However, with the actuation mechanism proposed by List et al., the programmed insertion into specific targets would not be possible. By introducing the toehold-mediated strand displacement locks an additional level of programmability was introduced, since a complementary sequence is needed to activate the locks in this setup.*

3. As hydrophobic tags, the authors use a mixture of cholesterol- and palmitoyl-functionalized oligos. What is the reasoning behind this? Have the authors quantified the benefits (less aggregation, better insertion...)? In the SI (Note S2), the authors discuss the free energy gain for pore insertion, but they just assume the same energy gain for cholesterol and palmitoyl insertion. This makes it even more unclear why they chose the mixture. This should be discussed.

- *This is a good point that we definitely have considered ourselves. We have not done a systematic study of different kinds of tags so we decided to leave any speculation out of the manuscript. The reason for using a mixture of functionalised oligos is two-fold. Firstly, in the early stage of the project 18 cholesterol anchors were ordered from IDT and later we took advantage of 28 palmitoyl anchor provided by our collaborator (see acknowledgements (<https://onlinelibrary.wiley.com/doi/full/10.1002/anie.201703243>)). As the project progressed we found the best results using the highest number of hydrophobic moieties, hence the 18+28. Secondly, from the literature it has been demonstrated that cholesterol- and palmitoyl-anchored DNA have different effects on the lipid membranes (doi:10.1039/C7OB01939D), with cholesterol creating more leaky liposomes and palmitoyl being more fusogenic (tight anchoring). Hence, we reasoned that by including both types of lipids in the nanopore structure we may get the best of both worlds. Regarding the use of the same energy gain – we have not been able to find measurements of solvation energies in the literature.*
- *We have updated NoteS2 and added a sentence stating that we got most efficient insertion with all lipids together (p.55 li.121-122) and (p.3 li.76-77)*

4. Why does the nanopore appear longer in TEM than the theoretically expected value (35nm vs. 32nm)? Is it due to the staining?

- *In our experience, doing conventional negative stain TEM flattens 3D DNA origami structures to a certain extent, a phenomenon especially pronounced for hollow structures. We have done some preliminary cryo-EM, with limited success so far, but quantification of length yields a measure of 31.6 nm very close to 32 nm as expected.*
- *A Supplementary figure S9 showing cryo-EM data has been added and the sentence has been rewritten in the manuscript (p.5 li. 81-83)*

5. p. 5, li. 83: “In addition, a thermal denaturation assay confirmed the high structural stability (Fig. S10)” It would be helpful if the authors could add the determined melting temperature to this statement (56°C). This value is quite typical for DNA origami structures.

- *This is a good point and we have added this to the main text. (p.5, li.86).*

6. The dimerization is a nice feature of the DNA origami nanopores. How many percent of the pores are dimers before and after the introduction of the sticky ends? This information should be added to the main text. Why did they not attempt to measure the gated transport between compartments? Are the insertion rates too low?

- *We did not quantify the TEM pictures but based on gel quantification about 15 % of the monomers (A or B) formed in separate mixtures are found in the dimer state after 30 mins (S12 lane 2 and 3) – Clarifying information has been added to the figure legend. In contrast > 70% of the nanopore is dimeric when A and B pores are mixed after 180 mins. A comment about this has been added to the main text (p.5, li.90-91).*
- *The suggestion about making a gated channel between two compartments is indeed an interesting point and this is something we are currently pursuing. However, many factors need to align and the completion of this work will be beyond the scope of the current manuscript.*

7. The quantification of the insertion characteristics is very valuable. However, the conclusions are only valid, if the authors are certain that their SUVs are unilamellar. Have the authors quantified the fraction of multilamellar vesicles? I am concerned that it might actually be quite large, since extrusion was performed with large filter sizes between 100 and 400nm and without freeze drying.

- *We fully agree with the reviewer that SUVs have to be unilamellar for conclusions made in the presented work to be valid. As a matter of fact, during preparation the SUVs were subjected to 10 cycles of flash-freezing and thawing to ensure a unilamellar structure. This methodology was shown*

recently by us (<https://doi.org/10.1038/nchembio.213>) and others (<https://doi.org/10.1073/pnas.0907354106>), using electron microscopy imaging, to yield a negligible (<5%) population of multilamellar vesicles.

- Acknowledging that this was not clearly explained in the original manuscript, we have added a section in Methods in “TIRF single particle assay” and “SUV formation for TEM” section clarifying this (p.15 l.317 and li.350) and also highlighted it in the main text (p. 7 li.111-112)
-

8. The authors state that they observed 86 DNA nanopore insertions over the course of 8h. How many insertions of alpha-hemolysin did they observe in the same time period?

- While the primary scope of the presented work is to quantify characteristics of the DNA nanopore structure, we agree that statistics regarding alpha hemolysin measurements were not clearly stated. Following the reviewer’s comment, we have revised Figure 3 and specified that we observed a total of 182 fully formed and leaking alpha hemolysin pores and 86 fully penetrated and leaking DNA nanopores. Regarding insertion of alpha hemolysin, it should be noted that this protein forms a heptameric structure, which only results in pores in the cellular membrane when fully assembled.
- In order to directly observe the insertion of each of the seven alpha hemolysin monomers, each monomer would have to be fluorescently labelled. Distinguishing between the heptameric form and the assembly of 6 monomers would not be trivial and is outside the scope of the paper.
- In addition, the final concentration of alpha hemolysin (heptamer) used in the assay was 45 nM whereas the maximal estimated final DNA nanopore concentration used in this assay was 3 nM (15x less). This means that the insertion efficiency of the structures cannot be directly compared in the experimental setting used here.

9. The authors look at “the smallest or the largest SUVs” (p. 7, li. 126) to study the mechanosensitivity of the pores. How small is small and how large is large and what is the difference in curvature? 100 nm vs. 400 nm? Can they estimate the difference in membrane tension?

- We thank the reviewer for noticing the lack of clarity in the figure. The original figure was based on a) quantification of size distribution of the SUVs that display pore formation for each of the alpha-hemolysin and Nanopore experiments as we described recently (<https://doi.org/10.1038/nchembio.213>) b) identifying the median of each SUV population for each experiment and c) dividing the SUV population in the median. SUVs with sizes above the median were referred to as the “large population” and SUVs below the median as the “small population”. Following the reviewer comments and based on the fact that the two medians are not identical due to preferential pore forming of DNA NP on larger SUV (see fig. 3 of the original submission), we have now provided a new figure where the SUV population of the two experiments are divided above and below 70nm in diameter. While the number of SUVs on either side of 70 nm are not identical the provided figure clearly confirms our earlier conclusions: The flow rate for the large and small size distribution for the DNA nanopores are overlapping (3f) and thus the flow rate for the DNA nanopore are independent of SUV sizes. The flow rate for the small and large SUVs the for alpha hemolysin pore (e), are dependent of the vesicle size and thereby the membrane tension. These results are confirmed with a two-tailed Kolmogorov-Smirnov test rejecting that the two distributions underlying are drawn from the same distribution with a P-value of $1.34 \cdot 10^{-6}$.
- For clarity we have updated the manuscript (p.7-8 li.135-137) with text according to the new figures and added the table below in the supporting information. We have also calculated the membrane tension using the method described in our recent paper (<https://doi.org/10.1038/nchembio.1733> or <https://doi.org/10.1016/j.bpj.2013.11.023>) and provided the new Supplementary table S1.
- New panels for figure 3ef
- e) f)

-
- **Additional Supplementary Table S1:**
- Liposomes sizes where DNA nanopore successfully form pores

Liposome selection	Diameter size [nm]*	Membrane tension [mN·m ⁻¹]*
Median	78.37 ± 35.91	48.73 ± 22.60
Max	139.30 ± 0.78	27.40 ± 1.87
Min	52.18 ± 0.28	73.25 ± 4.99

-
- Liposomes size information where Alpha hemolysin successfully forms pores.

Liposome selection	Diameter size [nm]*	Membrane tension [mN·m ⁻¹]*
Median	57.67 ± 10.26	66.25 ± 12.64
Max	92.89 ± 0.52	41.10 ± 2.80
Min	50.71 ± 0.27	75.38 ± 5.14

- * Provided errors is given as standard deviation
- Table x: Statistics on size distributions for both Alpha-Hemolysin and DNA Nanopore experiments.

10. The lack of mechanosensitivity of the DNA pores is a very interesting result, however it contradicts previously published work: MD simulations showed that the conductance of DNA nanopores depends on membrane tension (Yoo et al., J. Phys. Chem. Lett., <https://pubs.acs.org/doi/10.1021/acs.jpcclett.5b01964>) and experiments showed that the conductance states can be influenced by membrane tension (Seifert et al., ACS Nano, <https://pubs.acs.org/doi/10.1021/nn5039433>). These results have to be mentioned and discussed! Do the authors believe the double layer of DNA is the reason? It would be really good if they could test the mechanosensitivity of previous pore designs in a follow-up study.

- *Thanks for raising this point. Yes, we are fully aware of the fact that previously explored 6hb pores, show a “gated” behavior related to the tension and voltage applied. The design of our pore was indeed intended to avoid this by using a double layer. Thus, the results are not contradictory. We fully agree with the referee that the double layer of DNA is the most possible explanation for the mechano-insensitive efflux. We have added a comment in the revised manuscript discussing this difference (p.8, li.143-146).*
- *Indeed, the work and simulations of Yoo et al., display a correlation between lateral pressure and conductance. The pore used is a 6hb pore of arranged (ds)-DNA helices yielding a single-walled, and thus less rigid, pore compared to the double-layered pore presented here, which basically arranges a layer of 6hb pores to form the large inner pore. This element combined with the high surface tension (-100 dyn/cm = -0.1 J/m² as compared to 10⁻⁴ J/m² in our system) could be responsible for the reported differences between the studies. Similarly, the experiments of Seifert et al. are performed on a single-layered nanopore. We also wish to highlight that surface tension variation is suggested, albeit not measured directly, as a possible explanation for their observed differences between their methods.*

- *Another point we would like to make is that while the efflux was insensitive to membrane tension, the actual insertion of the DNA Nanopore is mechanosensitive. This was already highlighted in Supplementary Figs. S20 and S28 and discussed in the main text and we had highlighted it further in the revised version (p.8 li.151-154). We now also cite the above-mentioned studies and suggest that the double layer may be the underlying reason for this difference, explaining that future experiments are needed to verify this. (p.8 li. 143-146)*
- *The results presented in our study are thus in our eyes not a contradiction to previous work but rather an addition to the discussion that mechanosensitivity of the DNA pores, as mentioned by the referee, can be improved if using a more rigid structural scaffold.*
- *It will indeed be very interesting to study this point further in future studies.*

11. Osmotic pressure differences and ionic conditions in the buffer are crucial parameters governing the insertion properties of DNA pores. While the authors provide the buffer conditions for the formation of SUVs, GUVs and DNA origami individually, it is difficult to find out what final buffer was inside and outside the vesicles during the experiment. Providing this information explicitly will add value, especially for the quantification of the insertion characteristics.

- *Thank you for pointing this out - these factors are indeed very important, especially regarding the GUVs where we have carefully monitored and fitted the osmotic pressure prior experiments using an osmometer similarly to a related study (<https://www.nature.com/articles/ncomms12787>). Addition of the nanopore in the 500 mM 1xTAE buffer did not significantly change the osmometer readings of the outer solution used in the GUV measurements.*
- *The SUVs, on the other hand, are more robust and in addition the inside and outside buffers are largely the same as surface immobilization is done by neutravidin-biotin capture.*
- *The information has been added to the relevant method sections. (p.15, li.338-340)*

12. The authors state that they made SUVs between 100 and 400nm by extrusion and checked the size via DLS (see Materials and Methods). However, the DLS data is not provided. This data should be added to the SI, especially since extrusion with larger filters often leads to a deviation from the desired pore size.

- *Thanks for bringing up this point. Data for DLS measurements has been added to the SI (See new Supplementary Fig. S11).*
- *To be noted, in this section we mostly used the 200 nm PC membrane filters.*
- *Also, in this part of the study 10 Freeze and Thaw cycles were done before extrusion; this information has been added to the methods section. (p.15 l.317 and li.350)*

13. All previous DNA nanopores exhibited voltage gating, which somewhat limits their applicability for nanopore-based single-molecule detection. Do the authors believe that the use of a double layer of DNA would suppress voltage gating? This would be an immense advantage of the design. Ionic current recordings could provide the ultimate proof, but at least some discussion about this point should be added to the manuscript.

- *This is clearly an interesting point that we intend to pursue in the future. Due to the large lumen and the 6hb layer in our nanopore, differences in conductance may be too small to detect experimentally but this needs to be tested for a clear answer.*
- *A section has been added to the main text to discuss this. (p.8 li.143-146)*

14. p. 12, li. 210: For clarity, replace “lumen size” by “pore diameter”.

- *Lumen size has been changed to pore inner-diameter to stress the fact it refers to the porous part for the molecules to translocate through. (p.12 li. 230)*

Reviewer #2 (Remarks to the Author): **BLUE changes in paper**

The manuscript from Kjems and coworkers describes a synthetic DNA nanopore for the programmable and size-selective translocation of molecules through lipid bilayers.

The main novelty of this work relies on the functional versatility of the DNA nanopore structure. Differently from previously described DNA-based nanopores, this structure enables the size-selective (in and out) translocation of large or small molecules through modification of its lumen with PEG-ylated strands and their further removal by single-strand displacement mechanisms. In addition, the DNA nanopore inserts into lipid bilayers upon actuation by external triggers. These are DNA strands, either added in solution or present on the lipid surface, that bind to specific regions of the structure and operate by reconfiguring the nanopore from a closed to an open form, in which three flaps expose previously hidden lipid moieties. This structural change enables tight binding of the nanopore to the lipid membrane and avoids unwanted (and previously observed) hydrophobic-driven aggregation. The DNA nanopore and its functioning has been thoroughly characterized by TEM and smTIRF as well as by gel electrophoresis and FRET spectroscopy.

In general, the work is well written, clear and pleasant to read. I noticed only few **typos** (listed at the end of this report). I particularly liked the experiments described in figure 4 and 5. In my opinion, the controllable size-selective channeling properties of the DNA nanopore by plugging/unplugging of its lumen is the strongest point of this work and nicely shows how the smart combination of simple molecular mechanisms on robust scaffolds can be used to generate functional nanodevices with advanced properties. The data obtained are solid, consistent with the hypotheses and well-illustrated. I am therefore fully favorable to publication of this work in Nature Communication upon few minor alterations that I suggest below. The authors may consider to address my comments for improvement of the paper and a few other questions that could satisfy the curiosity of the interested reader.

We thank the reviewer for acknowledging the novelty of the work, the solid consistent data and for recommending publication to Nature methods after minor alterations. Below we have detailed our answers to all the reviewer comments.

1. Lines 59-65 describe the novelty and strength of the work. I like very much point III (actuation of the device); however, I found somehow difficult to grasp it from the present description in the main text or from figure 1. This point became clearer to me only after re-reading how the device works, which is reported few lines below. I would therefore suggest to describe first the two states of the device (i.e. the closed and open form), making also the mechanical transformation from one to the other clearer in the schematics of figure 1. In my opinion, this will emphasize the smart strategy employed by the authors to control membrane insertion and will guide the reader smoother to the next points.

- *Thanks for raising this point. We have tried to rephrase this part to enhance readability (p.3, li. 62-67).*

2. Which type of DNA nanopores has been used in the experiments illustrated in Figure 3a-g? maybe pre-opened L-? I find hard to guess it from the schematics in a). Can the authors please state it clearly in the main text and legend? What is the average size of a “small” SUV and a “large” SUV?

- *We have used the open and pre-lipidated form (L+) and have updated the manuscript and figure legend to illustrate this. (p.9 li.168-169)*
- *In relation to the second comment, please see our response to comment 9 of reviewer 1. In brief, in the original submission we had split SUV sizes at the median for each of the hemolysis and Nanopore experiments. This resulted in two equal populations but with different mean and median sizes, as hemolysis and DNA nanopore display preferential pore formation for different curvatures. In the revised figure 3e-f we have split the SUV population above and below 70nm. Median sizes are*

written in the figure and are discussed in the supporting Table1 but are also attached below for convenience. We note that while DNA nanopore and hemolysin are exposed to identical SUVs, the DNA nanopore displays a higher propensity to form pores in larger SUVs compared to hemolysin (see also answer to comment 9 of reviewer 1)

- Average size for SUVs where DNA nanopore forms pores:
- Large population (>70nm) = 98.23±23.01nm
- Small population (<70nm) = 61.56±4.87 nm

- Average size for SUVs where the Alpha hemolysin forms pores:
- Large population (>70nm) = 76.32±7.21nm
- Small population (<70nm) = 58.29±7.18 nm
-

3. It would be interesting to see whether the numerical relationship between the flow rate and the pore size follows some particular law of diffusion. Do the authors think it may be useful to gather some more insights into the DNA nanopore functioning?

- We fully agree that investigating whether the flow rate vs. the pore size follows some particular law would yield valuable insights to further understand efflux through a pore and the impact of the pore diameter.
- For the 1.4 nm wide alpha hemolysin pore we measured a flow rate of 0.014 ±0.010 l/s. For the 9.6 nm wide DNA nanopore we measured a flow rate of 0.098 ±0.087 l/s. Thus, the flow rate in the DNA nanopore is 7.0 ±8.1 times larger (Error bar corresponds to one standard deviation). The pore diameter of DNA nanopore is 6.8-fold larger than the alpha hemolysin (Song, et. al., Science, <http://doi.org/10.1126/science.274.5294.1859>).
- The diffusion across the pore could be approximated with a flow that follows the Hagen-Poiseuille relation. The relation between pore size and flow can be described by the pressure difference in a Newtonian and incompressible fluid under laminar flow through a pore with constant cross section. In this case the flux is given as

$$J = \frac{\Delta p d_p^2 \varepsilon}{32 x_p \mu_p}$$

- Where Δp is the transmembrane pressure, d_p is the diameter of the pores, ε is the surface porosity of the liposome membrane, x_p is the length of the pore, and μ_p is the viscosity of the permeating fluid.

Assuming Δp , ε and μ_p are unchanged from the two experiments we get re relation:

$$\frac{d_{NP}^2}{x_{NP}} = const. \cdot \frac{d_{\alpha}^2}{x_{\alpha}}$$

$$\frac{(9.6 \cdot 10^{-9})^2}{35 \cdot 10^{-9}} = const. \cdot \frac{(14 \cdot 10^{-10})^2}{100 \cdot 10^{-10}}$$

$$\rightarrow const = \mathbf{13.43}$$

- The theoretical 13.4-fold increase for the ATTO 655 flow though the DNA nanopore is therefore in relatively good agreement within the experimentally observed flow increase, despite the fact that the observed flow rate could also be affected by the different pore charges, which is not taken into consideration in the theoretical calculation.
- This information is now added in SI (note S3) and discussed in the main text (p.7 li.132-134).
-

-

4. Line 152. "...suggesting a tethering mechanism independent of flap activation for lipidated structures". From the FRET experiments of Fig. S12, it looks like the lipidated structures are partially opened even in the "closed" configuration, which would actually well explain the high insertion yield of L- constructs. Can the authors shortly comment on that?

- *This is an interesting observation (that the FRET is slightly lower is lower in the lipidated state). However, in the absence of single-molecule FRET we are not able to distinguish whether this is due to partially opened structures, higher ratio of open structures or alterations in spacing distance or dipolar moment.*

5. What is the exact difference between Figure 3h and 3i? Does it mean that there could be docking without dye efflux?

- *We appreciate the reviewer observing this lack of clarity within the figure and have refined the legend to fully elucidate the difference.*
- *In figure 3h we look at the percentage of docking (how many of the observed SUVs are docked with nanopores). In figure 3i we extract the traces over time for the already docked SUVs to observe whether the docking leads to pore reorientation followed by insertion/flux of molecules.*
- *Indeed, there is a fraction of DNA nanopores that dock without forming a pore within the experimental time frame used (less than 10 % of docked DNA nanopores have fully penetrated the SUV and thus exhibit efflux in the experimental time frame). As such 3i indeed represents a subpopulation of 3h. We have clarified this further in the main text (Lines 197) as well as the figure legend*
- *We have modified the legend to Figure 3 to enhance the readability (p.6, Figure 3 legend li. 11-12).*

6. Experiments of Figure 4: did the authors check whether the obtained results of flow rate vs. particle size are in line with the expected values from the Stokes-Einstein equation on molecular diffusion? In general, I believe that by quantifying the observed effect in relation to a known physical law would give a stronger impact to this nice work and would possibly help to gain additional information on the performance of the synthetic device (see also point 3 above).

- *We thank the reviewer for noticing this, as comparing the observed flow rates with theory augments the clarity of the results. In brief our theoretical calculations of efflux across the membranes when compared with the experimentally observed values, indicate some form of hindrance.*
- *The diffusion of a dye molecule outside of the vesicle entails a) finding the pore following free diffusion and b) translocating across the pore following free diffusion and any potential mechanical hindrance.*
- *The Stokes-Einstein equation describes diffusive processes of spherical particles in liquid that undergo Brownian motion, but it does not adequately describe the highly restricted motion within the pore nor the repulsive interactions of similarly charged species. So we note, it can only be used as an approximation.*
- *The unrestricted diffusion of a particle is given by*

$$D = \frac{k \cdot T}{6 \cdot \pi \cdot R_H \cdot \eta}$$

- Where R_H is the hydrodynamic radius of the translocated molecule (here ATTO 655 and a 40 kDa Dextran-tetramethyl rhodamine).
- The root mean square displacement of the diffusive dye molecules within a time t in the 3 dimensions of the vesicle is given as:

$$\langle r^2 \rangle = 6Dt$$

- The flow rate constant k and the half-life time τ are inverse correlated. This gives that the diffusion coefficient and the rate constant are proportional:

$$k \propto \frac{1}{\tau} \propto \frac{D}{r^2}$$

where r is the radius of the vesicle.

The half-life time for an ATTO 655 dye encapsulated within a vesicle with a radius of 35nm would thereby theoretically be expected to be $\tau \cong 3.3 \cdot 10^{-6}$ sec. We would therefore expect that within microseconds, half of the dye molecules have searched the vesicle and found the pore. The unhindered diffusion out of the pore is expected to occur in the same time scale. Our direct experimental observation of rates however shows efflux rates in the order of milliseconds to seconds (dTMR-40k translocating through the DNA nanopore of: 0.0057 ± 0.0036 l/s, while ATTO 655 flow rate was found to be 0.098 ± 0.087 l/s).

An explanation for this observed reduced translocation could be that the translocated molecules interact electrostatically with the pore or interact with the inner membrane of the vesicle, both of which could result in hindrance and therefore a delayed translocation.

- Because we have used the same pore and membrane composition, we can assume that the effect of Pore and membrane characteristic are similar for the Atto-655 and dTMR-40k the and the relation between the observed and theoretical flow rates and pore size can be analyzed. Comparison of flow rates yield a 17.09 ± 18.50 -fold faster flow rate for translocation of the small ATTO 655 molecule (error bars correspond to one standard deviation). We note that the temporal resolution of the confocal experiments limits the accurate extraction of flow rates.
- Using the Stokes-Einstein Kinetics we would expect:

$$D_{ATTO} = const \cdot D_{dextran}$$

$$\frac{k \cdot T}{6 \cdot \pi \cdot R_{H \text{ ATTO}} \cdot \eta} = const \cdot \frac{k \cdot T}{6 \cdot \pi \cdot R_{H \text{ Dextran}} \cdot \eta}$$

$$\frac{1}{R_{H \text{ ATTO}}} = const \cdot \frac{1}{R_{H \text{ Dextran}}}$$

- const = **7.59** times higher flow rate for the ATTO 655, using a hydrodynamic radius of 5.86 Å for ATTO 655 and 44.5 Å for the 40 kDa Dextran-tetramethyl rhodamine. While Stoke-Einstein does not include the highly restricted motion within the pore, the observed hindrance, or the repulsive interactions of the two dyes molecule species, the calculated rates relation is in relatively good agreement with the recorded values.
- We have added this section in the supplementary information (note S4) and also added a discussion in the main text. (p.12 li.217-219)

7. Experiments described in Figure 4 and 5. Plugging experiments were performed to monitor the different influx rates of dyes through GUV-tethered nanopores, whereas unplugging experiments were performed to monitor the efflux of previously GUV-internalized dyes. Is there a practical reason for this experimental choice? Do the authors believe that the performance of the device is affected by the direction of dye flux? That is, are influx and efflux rates identical for the same molecule through the same DNA nanopore?

- *The reason for performing the unplugging experiment on the TIRF platform was mainly that we needed to add “unplugging” strands. In the GUV setup we relied on the heavy GUVs to settle on the surface by gravity, while the SUVs were tethered using biotin-neutravidin. Thus, if we added “unplugging” strands in the GUV setup, we observed movement of the liposome and loss of tracking. In addition, the GUVs had a limited temporal resolution at our CLSM setup of 10 min while we could achieve sub-5-second resolution and more SUVs in a field of view with the TIRF setup.*
- *Regarding the performance and direction of the dyes. This is a very interesting point indeed. Generally, the rate is defined by the difference in chemical potential as described by Ficks 1st and 2nd law. By assuming the charge-effect of the pore is identical inside and outside, that the curved bilayers potential is not contributing, and the chemical potential difference is the same between the SUV and GUV experiment, identical initial rates would be expected.*
- *As GUVs are bigger, a larger chemical potential will be maintained for longer periods.*
- *Eventually, the efflux is expected to be faster than the influx due to the “infinite” outer volume, especially for the SUVs.*

Some comments on the Suppl. Information:

In general, at some points of the SI, there seems to be a formatting problem (some images are moved away from the legend and/or only partly visible).

- *Thanks for bringing this up. This must be a formatting issue from uploading the file as this is not seen in our version. We will make sure the resubmitted manuscript will be of satisfactory quality.*

8. The analysis of structural design is very interesting and will surely be of help for the development of the DNA nanotechnology field. Can the authors state which criteria must be fulfilled to have what they describe as a structural “valley”?

- *The staple strand analysis looks at the melting temperature (or number of basepairs) for each hybridization domain for each staple strand. If we look at Rothmund’s original origami designs from his 2006 paper, the staple strands in his design generally have three hybridization domains: One long domain (16 nt) in the middle of the staple strand and two smaller domains (8 nt) at each end of the staple strand. During annealing, the long domain will hybridize to the scaffold first, while the two smaller domains subsequently hybridize to other regions of the scaffold. Now plot the domains of the staple strand on the x-axis (domain 1, 2, and 3), ordered sequentially from 5’-end to 3’-end, with the number of base-pairs for each domain, on the y-axis. (*) Domain 1 has 8 basepairs, so $(x, y) = (1, 8)$. Domain 2 has 16 basepairs, so $(x, y) = (2, 16)$. Domain 3 has 8 basepairs, so $(x, y) = (3, 8)$. If we connect the points, they form a concave down graph (“mountain” shaped). Having a concave down staplestrand domain graph like this is important, as it ensures that the staple strands does not get topologically “trapped” during annealing. Compare this to a staple strand with two long domains at the ends, and one or two shorter domains in the middle. For instance, if the 5’-end domain 1 has 16 bp, so $(x, y) = (1, 16)$, domain 2 has 8 bp, so $(x, y) = (2, 8)$, and domain 3 has 16 bp, so $(3, 16)$. Plotting these points gives us a concave up graph (“valley” shaped). During annealing, the two long domains at the end of this staple strand will anneal first. But now, with the ends fixed, the middle domain is not free to rotate, so it will not be able to assume the correct topological configuration (i.e. winding $\frac{3}{4}$ of a turn around the scaffold).*

- When performing the “valley score” analysis, we look at how many “valleys” each strand has and plot that as a histogram. The optimal design, by this measure, is one where all staple strands have zero valleys.
- This is just one of two automated analyses that we employ to evaluate how “good” the staple strands are in a DNA origami design, and to catch undesired mistakes before ordering staple strands. We mostly use it as an internal tool to optimize a given DNA origami design, as we iterate through different variants of a design.
- (*) This “valley score” procedure can also be done more accurately when the sequence is known, plotting on the y-axis the domain’s melting temperature, T_m , instead of using the number of base-pairs, bp.
- The analysis process has been implemented in the described caDNAno plugin, so it can be used directly from within caDNAno during the design process.

9. Also, the note on the free energy of insertion is very interesting. Here, I would suggest to rephrase the second sentence (apparently a typos) and define the values of “gamma”L and “gamma”S in the equation (probably the bilayer tension due to different contributions). The referred equation 4.2, for calculation of the number of hydrophobic tags needed, is not reported. It would be nice to have it directly in the SI to help the reader to grasp the message without recurring to literature search. A final point that the authors may consider to address is the energetic contribution given by palmitoyl when compared to cholesterol. Is there a particular reason why they decided to use 28/46 palmitoyl and 18/46 cholesterol moieties for DNA nanopore modification? As they are chemically very different (long hydrophobic chain vs. steroid), one would imagine that their energetic contribution to insertion is also different.

- Thanks for the points regarding the description. Changes have been applied to guide the reader (p.S4, li.101-102 + 107-111).
- Regarding the choice of lipid moieties, this point was also raised reviewer 1 and we will give the same answer here
- This is a good point and the latter part is something we definitely have considered a lot ourselves. The reason for using this mixture of lipids is two-fold.
- Firstly, the palmitoyl anchor was accessible to us from collaborators (see acknowledgements) where we have had other collaborations before with the benefit of using a similar (double)palmitoyl anchor (<https://onlinelibrary.wiley.com/doi/full/10.1002/anie.201703243>). In addition, the 18 cholesterol anchors were ordered by IDT in an early phase of the project to test if lower amounts were needed. As the project progressed, we found the best results using the highest number of hydrophobic moieties hence the 18+28.
- Secondly, from the literature cholesterol and palmitoyl anchored DNA have been shown to have different effects on the lipid membranes (doi:10.1039/C7OB01939D), with cholesterol creating more leaky liposomes and palmitoyl more fusogenic (tight anchoring), thus the inclusion of both is an attempt to get the best of both worlds.
- In continuation, no we have not done a deep analysis of the effect yet, but this is definitely an interesting point and could be a follow-up paper in itself.
- Regarding using the same energy gain – this is due to lack of good solvation energies and measures in the literature, thus it is only meant to provide a concept of the needs.
- Indeed, the palmitoyl and cholesterol are very different, and their energetic contribution is for sure different. We have, however, not been able to find good estimates of the solvation energy for palmitoyl hence the discussion is meant to serve as a concept of how to think about lipidation of large nanopores. It would be very interesting to do the same analysis with a much better estimate of the free energy gain. (To clarify we have underlined our assumptions and experiences in the discussion of note S2 (p. S4 li. 117-122))

10. Fig. S20, legend: I believe left and right of panel a) have been exchanged.

- *Thanks for pointing this out. Indeed, they were exchanged.*

Some typos and few minor comments:

11. Line 62: “the design permitS”.

- *Thanks – corrected.*

12. Figure 2d, lower: I am not able to see well the dimerized channel building a bridge between two vesicles. As this construct was not further analyzed in this paper, I would suggest the authors to consider moving this TEM image to the SI.

- *Thanks for bringing this up. The figure 2d lower image is meant to show how the two ends both interact with liposomes. While one end seems to be inserted, the other end interacts with a liposome using its one flap structure. This is indeed not a continuous channel but it demonstrates how both ends are “active” in liposome engagement.*

13. Line 106: PLL-PEG, please expand abbreviation for PLL

- *A full name has been added here. (p.6-7 li.109-110)*

14. Line 112: “..the dye efflux rate caused by pore formation was quantified”. Is there the possibility that the intensity of the loaded dye may decrease by photobleaching? Did the authors check for that?

- *We fully agree with the importance of checking for potential photobleaching, as we are extracting exponentially decreasing rates from the intensity trajectories. We have of course optimized imaging conditions and practically eliminated ATTO 655 bleaching within the experimental time frame. We realized that this was not obvious enough from fig 3c and 5b so we have added an extra figure in the supporting information (Figure S27) where we display 5 trajectories of non-leaking SUVs over a full experiment (approx. 9.6 h), showing no significant photobleaching. To further clarify, we also include a figure containing both a non-leaking SUV and a SUV undergoing efflux of ATTO 655 through a fully inserted DNA nanopore, displaying a major difference between the massive loss of intensity from ATTO 655 translocation compared to the negligible decrease from photobleaching.*
- *The figure is attached below, as well as added in the revised supplementary information Fig. S27 and discussed in the main text in the assay section line 123-125.*

- Figure xx: The figure shows intensity traces for ATTO 655 loaded SUVs recorded over a time period of 9.6 h with a temporal resolution of 4.6 s. (a) The photobleaching under these imaging conditions is negligible. (b) Comparing the stable non-leaking SUVs (black) to the DNA nanopore-induced leaking SUVs (red) confirms the negligible effect of photobleaching in our results.

15. Line 114: “ From ...10,000 liposomes a total of 86 nanopore insertion events was observed”. The reader would conclude that the yield of insertion is <1%. From the TEM images reported in the main manuscript and SI, it seems to me that the yield of insertion events is much higher. Very often one SUV is bound to many DNA nanopores. How do the authors comment on that?

- *This is a very good point and one of the main challenges for DNA nanopores. The insertion yield observed in well-hydrated liposomes is very low indeed. This low efficiency is probably explained by two main factors. The energetic barrier from introducing the DNA into a lipid bilayer and the relatively low concentrations used compared to those used for protein pores (mainly due to costs).*
- *We also note that ~10% of the docked nanopores properly insert and form pores on liposomes within the experimental time frame as shown in Fig. 3h-i*
- *During the TIRF measurement a certain final volume is required (we flow in 200 ul containing approximately 1 pmol nanopore). Using TEM, the 1 pmol can be limited to just 15 ul including SUVs providing a much higher final concentration. Notably, higher insertion frequency could be achieved by lowering the SUV concentration (providing a higher excess of the nanopores). In short, many factors can influence insertion efficiency.*

16. Figure 3: in the legend it should be stated that the DNA nanopore is labelled with ATTO 488.

- *Added to figure 3. (p.6, Figure 3 legend li.1)*

17. Line 179: “... still allowed translocation of the small SRB dye out of the GUVs”. Do the authors mean instead “into the GUVs”? All experiments described in this section refer indeed to dye-influx.

- *Yes indeed. Thanks for catching this obvious mistake.*
- 18. Figure 5: the final part of the legend is missing.
 - *Thanks – This has been corrected.*
- 19. Line 128: “a-hemolysin appearS mechanosensitive ...”
 - *Thanks – corrected.*
- 20. Line 134: “DNA nanopore has a rigid channel, which allows ...”
 - *Thanks – corrected.*
- 21. Line 163: can the authors provide the estimated Rh for the SRB? this will help the reader to make a full comparison between flux rates and molecular size.
Rh for SRB is estimated to ~5Å (3 aromats and ethyl chains from side to side in flat configuration).
This has been added to the main text. (p.10 li.181)

** See Nature Research's author and referees' website at www.nature.com/authors for information about policies, services and author benefits

This email has been sent through the Springer Nature Tracking System NY-610A-NPG&MTS

Confidentiality Statement:

This e-mail is confidential and subject to copyright. Any unauthorised use or disclosure of its contents is prohibited. If you have received this email in error please notify our Manuscript Tracking System Helpdesk team at <http://platformsupport.nature.com> .

Details of the confidentiality and pre-publicity policy may be found here
<http://www.nature.com/authors/policies/confidentiality.html>

Privacy Policy | Update Profile

DISCLAIMER: This e-mail is confidential and should not be used by anyone who is not the original intended recipient. If you have received this e-mail in error please inform the sender and delete it from your mailbox or any other storage mechanism. Springer Nature Limited does not accept liability for any statements made which are clearly the sender's own and not expressly made on behalf of Springer Nature Ltd or one of their agents.

Please note that Springer Nature Limited and their agents and affiliates do not accept any responsibility for viruses or malware that may be contained in this e-mail or its attachments and it is your responsibility to scan the e-mail and attachments (if any).

Reviewers' Comments:

Reviewer #1:

Remarks to the Author:

The authors have done a considerable amount of work to improve the manuscript and to address the points that were raised during the reviewing process. I fully support the publication in Nature Communications. However, I would like to raise one final point:

It is very interesting and for me unexpected that the authors observed 86 insertions of the DNA nanopores, while 182 of the 15x less concentrated alpha-hemolysin pores inserted.

When I read the original version of the manuscript, I was under the assumption that the insertion rate of the DNA nanopores is extremely low. With the new information, it turns out that the insertion of the DNA nanopores is actually 7x more efficient than the insertion of alpha-hemolysin! This would be remarkable as DNA nanopores were always thought to have low insertion rates. It would be a missed opportunity not to discuss this point.

Reviewer #2:

Remarks to the Author:

The authors addressed all my points in a totally satisfying manner. I have no further comments and fully confirm my support to publish this very nice work in Nature Communication.

Point-by-point response.

Reviewer remarks

Reviewer #1 (Remarks to the Author):

- The authors have done a considerable amount of work to improve the manuscript and to address the points that were raised during the reviewing process. I fully support the publication in Nature Communications. However, I would like to raise one final point:
It is very interesting and for me unexpected that the authors observed 86 insertions of the DNA nanopores, while 182 of the 15x less concentrated alpha-hemolysin pores inserted.
When I read the original version of the manuscript, I was under the assumption that the insertion rate of the DNA nanopores is extremely low. With the new information, it turns out that the insertion of the DNA nanopores is actually 7x more efficient than the insertion of alpha-hemolysin! This would be remarkable as DNA nanopores were always thought to have low insertion rates. It would be a missed opportunity not to discuss this point.
 - *Thank you for the comment. Indeed, we observe 86 DNA nanopore insertions and 182 alpha hemolysin insertions using conditions where the alpha hemolysin is 15-fold more concentrated than the DNA nanopore. The direct numbers can, however, not be compared as the number of tracked liposomes differs significantly. To detect the 86 DNA nanopores insertions we observed a total of 4412 liposomes where 865 showed nanopore docking and 86 of those were perforated. For alpha-hemolysin control experiments, we tracked only 400 liposomes to obtain 182 alpha hemolysin traces. In this case we could not measure docking efficiency as the AH was not fluorescently tagged. Thus, for the DNA nanopore we observe that 20% of SUVs are docked, out of which 10% leak within the experimental time frame, resulting to a total insertion ratio of ~2 %. For alpha-hemolysin the insertion ratio is ~40 %, in line with the 15-fold higher concentration used. To clarify this, we have added the following sentence to the manuscript:*
 - *“Looking at the nanopore efficiency of insertion, about 20 % of tracked liposomes were docked with DNA nanopores, from which about 11 % was perforated during the experiment (Supplementary fig. 21b). Although docking of alpha hemolysin was not tracked, about 40% of liposomes were perforated by it. Considering alpha hemolysin was used in a 15x higher concentration an approximal equal perforation per pore was obtained.”*

Reviewer #2 (Remarks to the Author):

- The authors addressed all my points in a totally satisfying manner. I have no further comments and fully confirm my support to publish this very nice work in Nature Communication.